# FairGB: A Fair Granular-Ball Generation Method for Data Classification

Qifen Yang [1]  Yuhui Deng [1]  Jiande Huang [1]  Peng Zhou [1]  Xiwen Lu [1]  Lin Cui [1]

## Abstract

With the widespread application of data-driven classifiers in high-risk domains, group fairness has become an important research focus. However, most existing methods rely on model constraints or data reweighting, which may limit interpretability or distort the original data distribution. Granular-ball computing (GBC), as a structured and interpretable learning framework, provides a natural foundation for incorporating group fairness into data partitioning. Based on this insight, we propose a **Fair G**ranular-**B**all **G**eneration framework (FairGBG), which employs fair clustering to maintain balanced proportions of sensitive groups within each granular ball (GB), thereby enhancing within-ball group fairness. Theoretical analysis shows that FairGBG can preserve high GB purity while satisfying group fairness requirements. Furthermore, we introduce a **Fair G**ranular-**B**all-based data **F**air **C**lassification method (FairGBFC), which leverages fair GBs to improve classification fairness. Experiments on multiple benchmark datasets demonstrate that, compared with existing GB generation methods, FairGBG can generate locally fair GBs. Moreover, compared with state-of-the-art fairness-aware baselines, FairGBFC achieves a superior trade-off between accuracy and fairness, effectively mitigating bias while preserving high utility. Our code is publicly available at https://github.com/jamesandai/FairGB.

## 1. Introduction

Recent years have witnessed the rapid development of data-driven classifiers in critical areas such as recruitment, healthcare, and criminal justice (Chen et al., 2023), raising widespread concerns about the social risks they pose. In

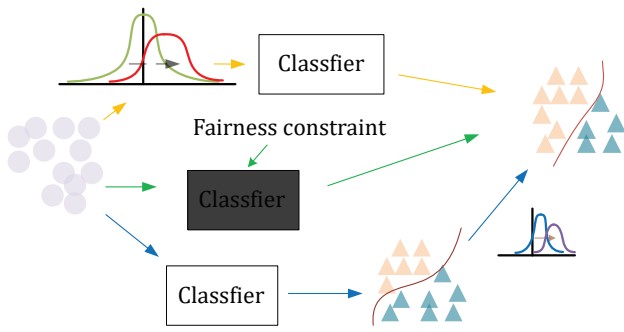

*Figure 1.* Brief illustration of existing fair classification methods, where the orange, green, and blue arrows indicate the processes of pre-processing, in-processing, and post-processing methods, respectively.

particular, unfair treatment of protected groups has emerged as a central issue in fairness research. Recent studies have shown that machine learning systems often exhibit systematic biases in their predictions toward certain population subgroups defined by sensitive attributes such as race, gender, and age, resulting in disproportionately negative impacts on these groups (Ledford, 2019; Bartlett et al., 2022; Zuo et al., 2023). As a result, promoting fairness and transparency within data-driven classifiers has become a prominent research focus in recent years (Ehyaei et al., 2024; Yang et al., 2026). As one of the first formalized concepts of fairness, group fairness focuses on ensuring equitable predictive outcomes across different demographic groups.

In recent years, researchers have proposed various methods to improve group fairness in machine learning models. These methods are typically categorized into three types based on the stage at which the intervention is applied: pre-processing, in-processing, and post-processing (Lassig & Herschel, 2025). The workflows of these methods are illustrated in Figure 1. Pre-processing methods (Jiang & Nachum, 2020; Lin et al., 2024; Shahbazi et al., 2024; Chan et al., 2024) mitigate data bias before training through techniques such as sample selection, data transformation, or reweighting. In-processing methods (Zhao et al., 2023; Baharlouei et al., 2024; Wang et al., 2025; Halim et al., 2025; Zhang et al., 2025b) incorporate fairness objectives or constraints directly into the model training process. Post-

[1]Department of Computer Science, Jinan University, Guangzhou, China. Correspondence to: Yuhui Deng <tyhdeng@jnu.edu.cn>.

*Proceedings of the 43rd International Conference on Machine Learning*, Seoul, South Korea. PMLR 306, 2026. Copyright 2026 by the author(s).

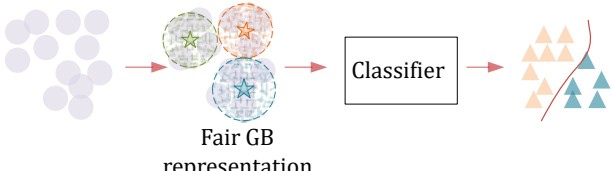

*Figure 2.* Brief illustration of our proposed method.

processing methods (Xian et al., 2023; Small et al., 2024; Zhang et al., 2025a; Denis et al., 2024) seek to improve fairness by adjusting the model's predictions after training, without altering the underlying model architecture. Despite recent progress, key challenges persist in fairness-aware classification. First, in-processing methods that enforce fairness during training often suffer from limited interpretability. Second, approaches relying on global-level interventions struggle to scale efficiently to large datasets. Finally, pre-processing and post-processing techniques, by modifying feature distributions or prediction outcomes, risk distorting the intrinsic data structure and compromising model utility.

To address the aforementioned challenges, this paper is motivated by a key theoretical insight: the objectives of maximizing granule purity and satisfying local fairness within granular balls can be effectively decoupled and jointly optimized through structural constraints. Based on this insight, we propose a **Fair Granular-Ball Generation** framework (FairGBG), which transforms the dataset into a set of high-purity and demographically balanced granular balls. The FairGBG framework proceeds in two stages: data augmentation and granular partitioning. First, to address inherent demographic imbalances, we utilize a Neighborhood Mixup strategy to augment minority groups, thereby achieving a balanced data distribution. Second, we propose the Group Aware Fair Clustering (GAFC) algorithm, which optimizes a fairness penalty to minimize group disparity in clustering. Through the recursive application of GAFC, FairGBG generates granular balls that maintain strict local demographic equilibrium. Building on these fair granular structures, we further propose a **Fair Granular-Ball-based Fair Classification** method (FairGBFC). FairGBFC establishes an adaptive mechanism that calibrates the influence of granular balls based on their purity and structural balance, thereby prioritizing reliable and fair regions for inference.

A brief overview of FairGBFC is shown in Figure 2. Building on the fair granular balls constructed by FairGBG, the framework inherently fosters group fairness in classification outcomes by representing the data space as a set of local, label-aware, and demographically balanced decision regions. Furthermore, it harnesses both the interpretability and the scalability of granular-ball computing. Specifically, each decision region is characterized by quantifiable structural statistics, such as purity, radius, center, and sensitive-group

balance, which enables FairGBFC to jointly consider classification reliability and local fairness during inference. In summary, the main contributions of this study are as follows:

- We propose the GAFC algorithm, which explicitly optimizes the trade-off between geometric compactness and demographic equilibrium. By enforcing strict demographic parity within the clustering process, GAFC provides a rigorous mathematical foundation for fair granular ball generation.

- We develop the FairGBG framework, utilizing recursive GAFC to construct high-quality granular balls, and theoretically establish that maximizing the purity of granular balls lowers the upper bound of unfairness within them.

- We introduce FairGBFC, a fair classification method that leverages the fair granular balls constructed by FairGBG to achieve fair inference via an adaptive decision mechanism. Extensive experiments demonstrate that our method significantly improves fairness while maintaining competitive classification performance.

## 2. Preliminaries

To facilitate understanding, we summarize the main mathematical notations and symbols used throughout this paper in Table 5 of Appendix A.

### 2.1. Demographic Parity

Demographic Parity (DP) is a widely used fairness criterion in classification tasks with a binary label $Y \in \{0, 1\}$. Its formal definition is given below.

**Definition 2.1** (Demographic Parity (Dwork et al., 2012))**.** Given input features $x \in \mathcal{X}$, a classifier $f : \mathbb{R}^d \to [0, 1]$ produces a predicted label $\hat{Y} = f(x)$. DP requires the prediction $\hat{Y}$ to be statistically independent of the sensitive attribute $S$. For multiple sensitive groups, i.e., $S \in \{1, \ldots, N_S\}$, where $N_S$ represents the number of sensitive attribute values, the metric generalizes to the maximum pairwise discrepancy between any two groups:

$$\Delta \text{DP} = \max_{u,v} \left| P(\hat{Y} = 1 \mid S = u) - P(\hat{Y} = 1 \mid S = v) \right|. \tag{1}$$

### 2.2. Granular-ball Generation

According to the GBC theory (Xia et al., 2019), the basic definition of a granular-ball (GB) is given below.

**Definition 2.2** (Granular-ball (Xia et al., 2019))**.** Let the dataset $D = \{(x_i, y_i)\}_{i=1}^{N}$ contain $N$ samples. The generated set of GBs is $G = \{gb_1, \ldots, gb_{N_{GB}}\}$, where each

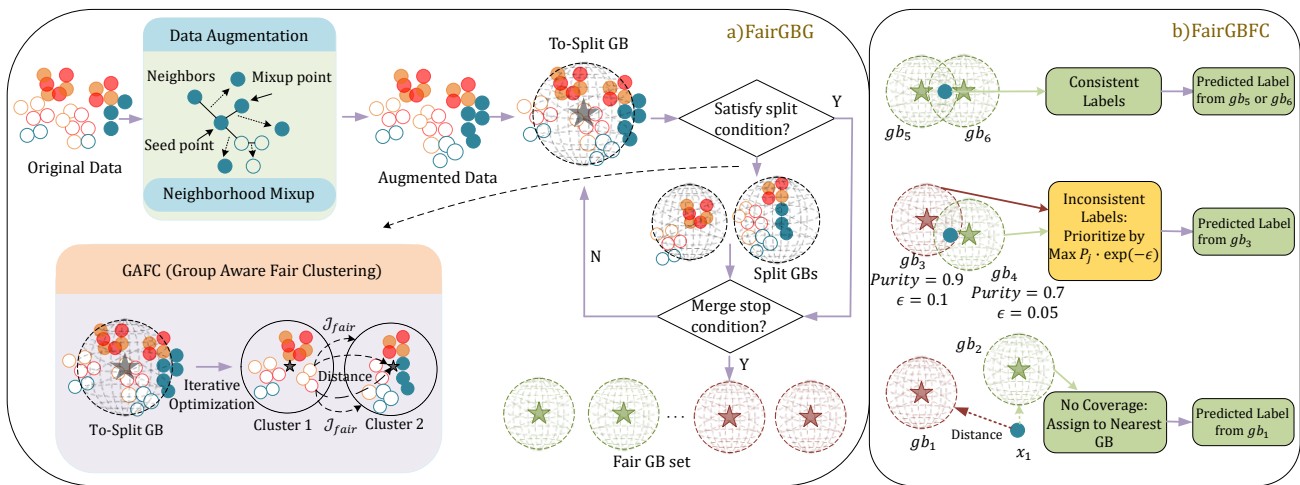

*Figure 3.* The framework of our proposed method. Solid and hollow circles represent positive and negative class samples, respectively, while different colors indicate sensitive attribute groups. a) FairGBG: The dataset is augmented and recursively partitioned into multi-granularity fair GB sets using the GAFC algorithm. b) FairGBFC: Leveraging the internal characteristics (i.e., purity and structural balance) of the generated fair GBs, this stage performs label assignment based on the granular coverage status of the test samples.

$gb_j$ contains $n_j$ samples denoted as $gb_j = \{(x_i, y_i) \mid i = 1, \ldots, n_j\}$. The key statistics are defined as follows: Center $O_j = \frac{1}{n_j} \sum_{i=1}^{n_j} x_i$; Radius $R_j = \frac{1}{n_j} \sum_{i=1}^{n_j} \|x_i - O_j\|_2$; Label $L_j = \arg\max_c |\{y_i = c \mid (x_i, y_i) \in gb_j\}|$; and Purity $P_j = \frac{1}{n_j} |\{y_i = L_j \mid (x_i, y_i) \in gb_j\}|$.

GB generation follows a top-down recursive strategy. Initially, the entire dataset is treated as a single coarse-grained GB. When a GB satisfies the splitting condition, it is further divided into sub-GBs. This recursion terminates when the splitting criteria are no longer met, yielding a final set of GBs characterized by their centers and radius.

## 3. Methodology

The overall framework, illustrated in Figure 3, operates in two sequential stages to ensure fairness. The motivation behind this design is to separate fair representation construction from fair decision-making: before performing classification, the original data space is first reorganized into locally fair and demographically balanced granular regions. Specifically, FairGBG (Figure 3a) focuses on representation learning by first applying data augmentation to balance the sensitive-group distribution and then recursively partitioning GBs with the proposed fair clustering algorithm. In this way, FairGBG constructs a fair multi-granularity space from the balanced data manifold, where each generated fair GB is expected to preserve both label purity and sensitive-group balance. Building on this fair granular structure, FairGBFC (Figure 3b) executes fair inference by exploiting the local properties of the generated fair GBs, such as purity and structural balance, to determine class labels.

### 3.1. Fair Granular Ball Generation (FairGBG)

We propose FairGBG, a hierarchical framework for generating fair GBs. As detailed in the left panel of Figure 3, this framework adopts a "coarse-to-fine" strategy to recursively partition the dataset. By explicitly enforcing demographic equilibrium within each GB during the generation process, FairGBG guarantees the local fairness of the resulting granular structures.

#### 3.1.1. DATA AUGMENTATION VIA NEIGHBORHOOD MIXUP

To rectify global demographic imbalances among sensitive groups without discarding original information, we employ the Neighborhood Mixup strategy (Liu et al., 2024b) prior to GB generation. By restricting interpolation to a seed sample and its $k$-nearest neighbors within the same sensitive group, Neighborhood Mixup helps preserve semantic coherence and avoids generating unrealistic samples that deviate from the local data manifold. For each under-represented group $u$, we randomly select a seed sample $(x_s, y_s, s_s)$ and identify its $k$-nearest neighbors within the same group. A target sample $(x_t, y_t, s_t)$ is then randomly sampled from this local neighborhood. Using a mixup coefficient $\alpha \sim \text{Uniform}(0, 1)$, a synthetic instance $(x^*, y^*, s^*)$ is generated. Numerical features are synthesized via linear interpolation: $x^*_{num} = \alpha x_{s,num} + (1 - \alpha)x_{t,num}$. Conversely, categorical features are determined by Bernoulli sampling, taking values from either the seed or the target with probability $\alpha$. This process expands the minority data manifolds until the group populations are globally equalized, i.e., $N_1 = \cdots = N_{N_S} = N/N_S$, thereby establishing

a rigorously balanced foundation for subsequent fairness optimization. Crucially, we strictly preserve the original positive label rate within each sensitive group during augmentation, ensuring that the intrinsic conditional label distributions remain unaltered while balancing group sizes.

### 3.1.2. GROUP AWARE FAIR CLUSTERING (GAFC)

The GAFC algorithm is designed to partition a given dataset into $K$ disjoint clusters $\{\mathcal{C}_1, \ldots, \mathcal{C}_K\}$ by optimizing for both geometric compactness and demographic equilibrium.

Let $\mathcal{S}$ denote the set of sensitive groups actually present in the input data, and let $m = |\mathcal{S}|$ represent the count of the existing groups. To formulate the conditions of group awareness, we define the Fairness Penalty Term $\mathcal{J}_{fair}$ based on the pairwise differences among these groups:

$$\mathcal{J}_{fair} = \sum_{j=1}^{K} \sum_{u \in \mathcal{S}} \sum_{v \in \mathcal{S}, v > u} (n_{j,u} - n_{j,v})^2, \quad (2)$$

where $n_{j,u} = \sum_{x_i \in \mathcal{C}_j} \mathbb{I}(s_i = u)$ denotes the number of samples belonging to sensitive group $u$ in cluster $\mathcal{C}_j$. GAFC generates fair clusters by minimizing the following objective function:

$$\min_{\{\mathcal{C}_j\}, \{O_j\}} \quad \sum_{j=1}^{K} \sum_{x_i \in \mathcal{C}_j} \|x_i - O_j\|^2 + \lambda \mathcal{J}_{fair}, \quad (3)$$

where $O_j$ is the center of cluster $\mathcal{C}_j$. By minimizing the pairwise squared difference of group counts, the algorithm penalizes clusters where any existing sensitive group disproportionately dominates, thereby driving each cluster toward a state of demographic equilibrium where $n_{j,u} \approx n_{j,v}$ for all $u, v \in \mathcal{S}$.

To solve this optimization, we leverage the algebraic identity that relates the pairwise variance to the sum of squares:

$$\sum_{\substack{u,v \in \mathcal{S} \\ u < v}} (n_{j,u} - n_{j,v})^2 \propto m \sum_{u \in \mathcal{S}} n_{j,u}^2 - \left( \sum_{u \in \mathcal{S}} n_{j,u} \right)^2. \quad (4)$$

Using this transformation, the gradient of the fairness term with respect to the count of a specific group $n_{j,u}$ can be derived as $\frac{\partial \mathcal{J}_{fair}}{\partial n_{j,u}} \propto m \cdot n_{j,u} - |\mathcal{C}_j|$, where $|\mathcal{C}_j|$ is the total cluster size. Let $V_{j,u}$ denote this gradient value, which quantifies the marginal cost of assigning a group $u$ sample to cluster $j$. The optimization proceeds via an iterative Expectation-Maximization scheme, where $t$ denotes the current iteration index:

**M-step:** Given the cluster assignments $\{\mathcal{C}_j^{(t)}\}$ from the previous iteration, we update the cluster parameters. The geometric centers are updated to the centroids of the current clusters: $O_j^{(t+1)} = \frac{1}{|\mathcal{C}_j^{(t)}|} \sum_{x_i \in \mathcal{C}_j^{(t)}} x_i$. Simultaneously, the

demographic potentials are updated to reflect the imbalance at state $t$: $V_{j,u}^{(t+1)} = m \cdot n_{j,u}^{(t)} - |\mathcal{C}_j^{(t)}|$.

**E-step:** With the updated centers $\{O_j^{(t+1)}\}$ and potentials $\{V_{j,u}^{(t+1)}\}$ fixed, each sample $x_i$ belonging to group $s_i$ is reassigned to the cluster $j^*$ that minimizes the combined cost:

$$j^* = \arg\min_j \left( \|x_i - O_j^{(t+1)}\|^2 + \lambda V_{j,s_i}^{(t+1)} \right). \quad (5)$$

This yields the new assignment set $\{\mathcal{C}_j^{(t+1)}\}$. Since the objective function is bounded below and the EM steps correspond to block coordinate descent on the transformed energy function, the algorithm is guaranteed to converge to a local minimum. Detailed proofs of convergence and time complexity analysis are provided in Appendix B. This fair clustering mechanism serves as the core splitting method for the subsequent Fair GB Generation process, where it is applied recursively to partition local data subsets.

### 3.1.3. RECURSIVE SPLITTING AND PRUNING MECHANISM

The FairGBG framework adopts a top-down recursive paradigm to partition the augmented dataset $\mathcal{D}_{aug}$ into a final set of fair GBs $G_{fair}$. The process initiates with the entire dataset as the root ball and iteratively evaluates each GB $gb_j$ in the queue based on its purity $P_j$ and sample size $n_j$. The procedure consists of the following phases:

- **Splitting Phase:** For a specific GB $gb_j$ identified as impure ($P_j < \tau$) yet containing sufficient samples ($n_j \geq N_S$, where $N_S$ denotes the total number of sensitive groups in the original dataset), we first compute a fairness penalty coefficient $\lambda_j$. This value is adaptively calibrated by scaling the pre-defined base $\lambda_{base}$ according to $gb_j$'s specific characteristics; specifically, $\lambda_j$ is proportional to the product of the ball's internal geometric magnitude and its demographic imbalance degree. Subsequently, we apply the GAFC algorithm ($K = 2$) to minimize Equation (3). Crucially, during this optimization, the generic parameter $\lambda$ is substituted with the specific $\lambda_j$, and the fairness term $\mathcal{J}_{fair}$ in Equation (2) is instantiated using only the set of sensitive groups $\mathcal{S}_j$ present within $gb_j$, thereby partitioning the ball into two child balls that maintain demographic balance among the existing groups.

- **Termination Phase:** The recursive subdivision terminates when all generated GBs satisfy the purity threshold ($P_j \geq \tau$) or cannot be further divided. To ensure robustness, only those satisfying both the purity criterion and the minimum size constraint ($n_j \geq N_S$) are retained in the final set $G_{fair}$.

The detailed algorithmic procedure is provided in Algorithm 2 in Appendix C.

### 3.1.4. RELATION BETWEEN PURITY AND GROUP FAIRNESS

Due to the lack of a unified definition of group fairness at the granular level, we formally define Label Demographic Parity (LDP) for a GB to quantify local fairness.

**Definition 3.1** (Label Demographic Parity). Label Demographic Parity requires that positive label rates be equal across all sensitive groups within a local region. For a GB $gb_j$, let $\mathcal{S}_j = \{u \mid n_{j,u} > 0\}$ denote the set of sensitive groups present, and let $p_{u,j} = P(Y = 1 \mid S = u, gb_j)$ be the positive label rate for group $u$. The quantitative metric is defined as the maximum pairwise deviation:

$$\Delta\text{LDP}(gb_j) = \max_{u,v \in \mathcal{S}_j} |p_{u,j} - p_{v,j}|. \tag{6}$$

A value of 0 indicates perfect local fairness across all present groups.

We demonstrate that when using GAFC to split GBs, the objective of increasing purity inherently aligns with enhancing group fairness, as formalized in Theorem 3.2.

**Theorem 3.2** (The Upper Bound of $\Delta$LDP in $gb_j$). *Consider a GB $gb_j$ with total sample size $n_j$. Let $m_j = |\mathcal{S}_j|$ be the number of sensitive groups present, and let $N_{min,j} = \min_{u \in \mathcal{S}_j} n_{j,u}$ be the size of the smallest group. Define the structural imbalance term as $\epsilon_j = n_j - m_j \cdot n_{min,j}$. For any $m_j \geq 2$, the $\Delta$LDP value is bounded by:*

$$\Delta LDP(gb_j) \leq m_j(1 - P_j) + \frac{\epsilon_j}{2N_{min,j}}, \tag{7}$$

*where $P_j$ is the purity of $gb_j$. In the ideal case of perfect structural balance (i.e., $\epsilon_j = 0$), the bound simplifies to $m_j(1 - P_j)$.*

The proof is provided in Appendix D. Theorem 3.2 elucidates the relationship between purity and LDP in GB: increasing purity ($P_j \to 1$) directly lowers the theoretical upper bound of unfairness. In the ideal scenario where a GB achieves unit purity ($P_j = 1$) and perfect structural balance ($\epsilon_j = 0$), the LDP value vanishes to 0, guaranteeing perfect local fairness. Crucially, the theorem implies that even when structural balance is imperfect ($\epsilon_j > 0$), increasing purity suppresses the unfairness bound. This motivates our GAFC algorithm, which simultaneously minimizes structural imbalance (reducing $\epsilon_j$) via the fairness penalty $\mathcal{J}_{fair}$ and enhances purity through recursive partitioning, ensuring the generated GBs serve as high-quality and fair representations.

### 3.2. Fair Granular Ball-based Fair Classification (FairGBFC)

Based on the constructed set of fair GBs $G_{fair} = \{gb_1, \ldots, gb_{N_{GB}}\}$, we propose FairGBFC for test sample classification. As illustrated in the right panel of Figure 3(b), unlike traditional approaches that depend on complex fairness constraints or post-hoc adjustments, FairGBFC directly leverages the intrinsic properties of the generated fair GBs, such as their purity and structural balance, thereby enabling fair classification.

To address overlaps between the GBs, we employ a prioritized hierarchical decision rule. Let $d(x, gb_j) = \max(\|x - O_j\|_2 - R_j, 0)$ denote the Euclidean distance from a query sample $x$ to a GB $gb_j$, and let $\mathcal{B}_x = \{gb_j \mid d(x, gb_j) = 0\}$ represent the set of GBs covering $x$. The assignment proceeds in two stages:

**1. Samples within GBs ($\mathcal{B}_x \neq \emptyset$):** If $x$ is covered by one or more balls that share a consistent label, $x$ directly inherits this label. However, if the labels of the covering balls are inconsistent, the label of the GB $gb_{j^*} \in \mathcal{B}_x$ is assigned to $x$, where $j^*$ is determined by:

$$j^* = \arg \max_{gb_j \in \mathcal{B}_x} (P_j \cdot \exp(-\epsilon_j)). \tag{8}$$

This mechanism ensures that in ambiguous regions, the decision is dominated by the GB that simultaneously demonstrates high classification reliability and superior structural balance.

**2. Samples outside GBs ($\mathcal{B}_x = \emptyset$):** If $x$ does not belong to any GB, we assign the label of the nearest GB to $x$.

We now establish a theoretical guarantee for FairGBFC by deriving an upper bound on the global unfairness based on the aggregated local structural imbalances of the GBs, as follows.

**Theorem 3.3** (Global Fairness Bound). *Assume the feature space is partitioned into $N_{GB}$ disjoint regions $\{\mathcal{R}_j\}$. Assume the dataset is globally balanced with $N_S$ sensitive groups, each of size $N/N_S$. The $\Delta DP$ of the predicted outcomes is bounded by:*

$$\Delta DP \leq \frac{N_S}{N} \sum_{j=1}^{N_{GB}} (\epsilon_j + (N_S - m_j)N_{min,j}). \tag{9}$$

*In the ideal case where all groups are present in every GB (i.e., $m_j = N_S$ for all $j$), the bound simplifies to $\frac{N_S}{N} \sum \epsilon_j$.*

The proof is provided in Appendix E. This theoretical insight provides a rigorous justification for the comprehensive design of our framework. Specifically, the Neighborhood Mixup strategy augments minority sensitive groups to maintain sensitive-group balance, thereby minimizing the missing group penalty (i.e., driving $m_j \to N_S$). Meanwhile,

*Table 1.* Comparison of FairGBFC and GB-based classification methods on binary sensitive attributes. Results include mean $\pm$ standard deviation. Best results are **bolded**, and second-best are underlined.

| Method | Approval (Gender) | | | | | Law (Gender) | | | | |
|---|---|---|---|---|---|---|---|---|---|---|
| | ACC↑ | F1↑ | ΔDP↓ | ΔEO↓ | Time | ACC↑ | F1↑ | ΔDP↓ | ΔEO↓ | Time |
| GBKNN | 0.835±0.029 | 0.807±0.043 | 0.101±0.036 | 0.110±0.060 | 0.270 | 0.945±0.001 | 0.971±0.001 | 0.010±0.008 | 0.061±0.052 | 1.890 |
| ACC-GBKNN | 0.839±0.034 | 0.821±0.037 | 0.085±0.043 | 0.105±0.096 | 0.139 | 0.945±0.001 | 0.971±0.001 | 0.008±0.005 | 0.028±0.013 | 4.215 |
| GB-KNN+ | 0.842±0.036 | 0.821±0.043 | 0.104±0.030 | 0.116±0.066 | **0.032** | 0.945±0.001 | 0.972±0.001 | 0.014±0.009 | 0.079±0.068 | 1.128 |
| FairGBFC | **0.864±0.018** | **0.844±0.019** | **0.062±0.019** | **0.037±0.013** | 0.068 | **0.949±0.000** | **0.974±0.000** | **0.000±0.000** | **0.000±0.000** | **0.004** |

| Method | Por (Sex) | | | | | German (Gender) | | | | |
|---|---|---|---|---|---|---|---|---|---|---|
| | ACC↑ | F1↑ | ΔDP↓ | ΔEO↓ | Time | ACC↑ | F1↑ | ΔDP↓ | ΔEO↓ | Time |
| GBKNN | 0.841±0.008 | 0.101±0.062 | 0.032±0.020 | 0.070±0.042 | 0.236 | 0.691±0.021 | 0.803±0.019 | 0.186±0.045 | 0.245±0.054 | 0.875 |
| ACC-GBKNN | 0.827±0.027 | 0.205±0.149 | 0.036±0.022 | 0.129±0.078 | 0.095 | 0.682±0.013 | 0.793±0.010 | 0.125±0.065 | 0.192±0.088 | 0.320 |
| GB-KNN+ | 0.809±0.022 | 0.304±0.050 | 0.094±0.027 | 0.149±0.057 | **0.033** | 0.628±0.019 | 0.740±0.015 | 0.121±0.049 | 0.200±0.010 | **0.075** |
| FairGBFC | **0.858±0.004** | **0.450±0.074** | **0.009±0.004** | **0.016±0.002** | 0.042 | **0.716±0.014** | **0.827±0.008** | **0.007±0.001** | **0.015±0.004** | 0.088 |

| Method | Adult (Sex) | | | | | Default (Sex) | | | | |
|---|---|---|---|---|---|---|---|---|---|---|
| | ACC↑ | F1↑ | ΔDP↓ | ΔEO↓ | Time | ACC↑ | F1↑ | ΔDP↓ | ΔEO↓ | Time |
| GBKNN | 0.803±0.003 | 0.551±0.008 | 0.186±0.010 | 0.160±0.036 | 20.40 | 0.793±0.003 | 0.874±0.001 | 0.039±0.011 | 0.054±0.022 | 26.538 |
| ACC-GBKNN | 0.799±0.003 | 0.577±0.006 | 0.218±0.011 | 0.168±0.035 | 8.879 | 0.790±0.005 | 0.874±0.003 | 0.050±0.004 | 0.058±0.012 | 15.511 |
| GB-KNN+ | 0.796±0.005 | 0.564±0.004 | 0.193±0.009 | 0.150±0.024 | 7.728 | 0.766±0.006 | 0.854±0.004 | 0.047±0.007 | 0.051±0.017 | 9.639 |
| FairGBFC | **0.812±0.005** | **0.594±0.026** | **0.089±0.027** | **0.083±0.031** | **4.215** | **0.810±0.005** | **0.889±0.003** | **0.018±0.013** | **0.029±0.011** | 7.265 |

the GAFC algorithm enforces intra-cluster equilibrium to suppress the structural imbalance $\epsilon_j$, while FairGBFC's adaptive decision mechanism further tightens this bound by down-weighting regions with high $\epsilon_j$. Consequently, Theorem 3.3 establishes a unified framework where both $\Delta$LDP and $\Delta$DP are strictly constrained by the same structural properties optimized by our method.

## 4. Experiments

This section presents experiments on six benchmark datasets to evaluate both components of the proposed framework. Specifically, FairGBG is evaluated by comparing the local fairness of its generated GBs with those of GB-based methods, while FairGBFC is compared with state-of-the-art GB-based and fairness-aware baselines to assess its classification utility and fairness.

### 4.1. Experimental Settings

*1) Datasets:* We conduct experiments on six real-world binary classification datasets that are widely used in fairness research: Approval (Quinlan, 1987), German (Dua & Graff, 2017), Por (Silva, 2008), Law (Wightman, 1998), Default (Yeh, 2009), and Adult (Dua & Graff, 2017). For each dataset, we choose one or two attributes related to ethics as sensitive attributes. Specific details regarding each dataset are provided in Appendix F.

*2) Baselines:* We compare the proposed methods with three state-of-the-art GB-based classification approaches, including GBKNN (Xia et al., 2019), ACC-GBKNN (Xia et al.,

2024a), and GB-KNN+ (Xie et al., 2024). In addition, to evaluate the effectiveness of our method in terms of classification fairness, we also compare it with five widely used fairness-aware methods, covering two pre-processing methods (GroupDebias (GDebias) (Chan et al., 2024) and Unbias (Jiang & Nachum, 2020)), two in-processing methods (CFA (Zhao et al., 2023) and $f$-FERM (Baharlouei et al., 2024)), and one post-processing method (DP_PP (Xian et al., 2023)). The compared fairness-aware methods are all implemented based on a multilayer perceptron.

*3) Experiment Protocol:* To ensure robust results and mitigate randomness, we employ a 5-fold cross-validation protocol, reporting the mean and standard deviation of the test scores across five independent runs. All baseline methods are implemented using the recommended parameter settings specified in their respective original papers. For our proposed FairGBFC, the key hyperparameters requiring configuration are the base fairness weight ($\lambda_{base}$) and the purity threshold ($\tau$). Detailed experimental configurations and hyperparameter settings for all compared methods are provided in Appendix G. To comprehensively assess performance, we utilize Accuracy (ACC) and F1 score (F1) as utility metrics. Simultaneously, we evaluate fairness using the Demographic Parity difference ($\Delta$DP) (Dwork et al., 2012) and the Equalized Odds difference ($\Delta$EO) (Hardt et al., 2016).

### 4.2. Comparison with GB-Based Classification Methods

Since the compared GB-based classification methods rely on the KNN approach, in addition to utility and fairness

*Table 2.* Comparison of local fairness in generated GBs measured by average ΔLDP. Lower values indicate better local fairness.

| Method | Approval (Gender) | Law (Gender) | Por (Sex) | German (Gender) | Adult (Sex) | Default (Sex) |
|---|---|---|---|---|---|---|
| GBKNN | 0.033 | 0.193 | 0.259 | 0.325 | 0.062 | — |
| ACC-GBKNN | 0.137 | 0.012 | 0.256 | 0.371 | 0.147 | — |
| GB-KNN+ | 0.137 | 0.012 | 0.121 | 0.115 | 0.121 | — |
| FairGBG | **0.001** | **0.000** | **0.006** | **0.003** | **0.013** | **0.008** |

*Table 3.* Comparison of different fairness-aware methods on binary sensitive attributes.

| Dataset (Task) | Method | Utility | | Fairness | |
|---|---|---|---|---|---|
| | | ACC↑ | F1↑ | DP↓ | EO↓ |
| Approval (Gender) | CFA | $0.823 \pm 0.052$ | $0.823 \pm 0.044$ | $0.082 \pm 0.030$ | $0.095 \pm 0.057$ |
| | $f$-FERM | $0.848 \pm 0.031$ | $0.829 \pm 0.035$ | $0.074 \pm 0.043$ | $0.089 \pm 0.043$ |
| | Unbias | $0.838 \pm 0.030$ | $0.812 \pm 0.042$ | $0.074 \pm 0.045$ | $0.079 \pm 0.060$ |
| | GDebias | $0.846 \pm 0.025$ | $0.818 \pm 0.033$ | $0.091 \pm 0.053$ | $0.097 \pm 0.075$ |
| | DP_PP | $0.845 \pm 0.023$ | $0.829 \pm 0.029$ | $0.113 \pm 0.079$ | $0.127 \pm 0.093$ |
| | FairGBFC | $\mathbf{0.864 \pm 0.018}$ | $\mathbf{0.844 \pm 0.019}$ | $\mathbf{0.062 \pm 0.019}$ | $\mathbf{0.037 \pm 0.013}$ |
| Law (Gender) | CFA | $0.944 \pm 0.002$ | $0.970 \pm 0.002$ | $\mathbf{0.000 \pm 0.000}$ | $\mathbf{0.000 \pm 0.000}$ |
| | $f$-FERM | $0.942 \pm 0.004$ | $0.968 \pm 0.005$ | $0.002 \pm 0.002$ | $0.016 \pm 0.013$ |
| | Unbias | $0.945 \pm 0.001$ | $0.972 \pm 0.001$ | $0.001 \pm 0.001$ | $0.010 \pm 0.006$ |
| | GDebias | $0.940 \pm 0.004$ | $0.970 \pm 0.002$ | $0.004 \pm 0.001$ | $0.039 \pm 0.018$ |
| | DP_PP | $\mathbf{0.949 \pm 0.000}$ | $0.974 \pm 0.000$ | $0.002 \pm 0.002$ | $0.021 \pm 0.019$ |
| | FairGBFC | $\mathbf{0.949 \pm 0.000}$ | $\mathbf{0.974 \pm 0.000}$ | $\mathbf{0.000 \pm 0.000}$ | $\mathbf{0.000 \pm 0.000}$ |
| Por (Sex) | CFA | $0.846 \pm 0.000$ | $0.000 \pm 0.000$ | — | — |
| | $f$-FERM | $\mathbf{0.878 \pm 0.017}$ | $\mathbf{0.558 \pm 0.118}$ | $0.101 \pm 0.044$ | $0.193 \pm 0.134$ |
| | Unbias | $0.846 \pm 0.000$ | $0.000 \pm 0.000$ | — | — |
| | GDebias | $0.877 \pm 0.025$ | $0.496 \pm 0.207$ | $0.056 \pm 0.057$ | $0.205 \pm 0.169$ |
| | DP_PP | $0.826 \pm 0.016$ | $0.139 \pm 0.115$ | $0.024 \pm 0.021$ | $0.157 \pm 0.140$ |
| | FairGBFC | $0.858 \pm 0.014$ | $0.450 \pm 0.074$ | $\mathbf{0.009 \pm 0.004}$ | $\mathbf{0.016 \pm 0.002}$ |
| German (Gender) | CFA | $0.705 \pm 0.010$ | $0.825 \pm 0.002$ | $0.014 \pm 0.028$ | $0.034 \pm 0.068$ |
| | $f$-FERM | $0.694 \pm 0.032$ | $0.787 \pm 0.024$ | $0.129 \pm 0.058$ | $0.201 \pm 0.120$ |
| | Unbias | $0.705 \pm 0.013$ | $0.823 \pm 0.005$ | $0.019 \pm 0.019$ | $0.055 \pm 0.051$ |
| | GDebias | $0.709 \pm 0.019$ | $0.820 \pm 0.012$ | $0.069 \pm 0.027$ | $0.153 \pm 0.047$ |
| | DP_PP | $0.701 \pm 0.005$ | $0.823 \pm 0.002$ | $0.034 \pm 0.045$ | $0.057 \pm 0.076$ |
| | FairGBFC | $\mathbf{0.716 \pm 0.014}$ | $\mathbf{0.827 \pm 0.008}$ | $\mathbf{0.007 \pm 0.001}$ | $\mathbf{0.015 \pm 0.005}$ |
| Adult (Sex) | CFA | $0.762 \pm 0.002$ | $0.000 \pm 0.000$ | — | — |
| | $f$-FERM | $0.805 \pm 0.002$ | $0.575 \pm 0.010$ | $0.197 \pm 0.009$ | $0.144 \pm 0.015$ |
| | Unbias | $0.790 \pm 0.006$ | $0.425 \pm 0.057$ | $0.233 \pm 0.020$ | $0.357 \pm 0.089$ |
| | GDebias | $0.804 \pm 0.004$ | $0.591 \pm 0.013$ | $0.097 \pm 0.023$ | $0.097 \pm 0.013$ |
| | DP_PP | $0.803 \pm 0.003$ | $0.547 \pm 0.030$ | $0.166 \pm 0.035$ | $0.121 \pm 0.042$ |
| | FairGBFC | $\mathbf{0.812 \pm 0.005}$ | $\mathbf{0.594 \pm 0.026}$ | $\mathbf{0.089 \pm 0.027}$ | $\mathbf{0.083 \pm 0.031}$ |
| Default (Sex) | CFA | $0.786 \pm 0.010$ | $0.879 \pm 0.004$ | $0.023 \pm 0.016$ | $0.030 \pm 0.037$ |
| | $f$-FERM | $0.802 \pm 0.004$ | $0.872 \pm 0.002$ | $0.036 \pm 0.008$ | $0.036 \pm 0.015$ |
| | Unbias | $0.809 \pm 0.004$ | $0.881 \pm 0.002$ | $0.024 \pm 0.009$ | $0.035 \pm 0.028$ |
| | GDebias | $0.809 \pm 0.003$ | $0.881 \pm 0.002$ | $0.019 \pm 0.012$ | $0.040 \pm 0.006$ |
| | DP_PP | $0.808 \pm 0.004$ | $0.884 \pm 0.002$ | $0.036 \pm 0.008$ | $0.042 \pm 0.023$ |
| | FairGBFC | $\mathbf{0.810 \pm 0.005}$ | $\mathbf{0.889 \pm 0.003}$ | $\mathbf{0.018 \pm 0.013}$ | $\mathbf{0.029 \pm 0.011}$ |

metrics, we evaluate the Average Runtime to assess computational efficiency. The corresponding results are shown in Table 1. Experimental results demonstrate that FairG-BFC consistently achieves not only the highest classification utility but also the best fairness performance across all datasets. Specifically, on the German dataset, FairGBFC improves ACC and F1 by approximately 3.6% and 3.0% respectively compared with the best-performing baseline, while significantly reducing △DP and △EO by 94.2% and 92.2% relative to the runner-up methods. Similarly, on the large-scale Adult dataset, our method increases ACC by roughly 1.1% while decreasing △DP and △EO by 52.2% and 44.7%, respectively. Regarding efficiency, FairGBFC exhibits excellent scalability; notably on Adult, it achieves the fastest inference time (4.215s), confirming that the proposed FairGBFC enhances equity with negligible or even reduced computational overhead.

### 4.3. Evaluation of Local Fairness in Generated GBs

To verify the effectiveness of FairGBG in generating locally fair GBs, we compare the granular structures generated by FairGBG with those produced by three GB-based classification methods. We use $\Delta \mathrm{LDP}(gb_j)$ to evaluate the local fairness of the generated GBs. Specifically, for each generated GB, $\Delta \mathrm{LDP}(gb_j)$ measures the maximum difference in positive label rates across sensitive groups within the same GB. A lower value indicates better local fairness, and a value close to zero suggests that the corresponding GB maintains nearly balanced label distributions among sensitive groups. Since multiple GBs are generated for each dataset, we report the average $\Delta \mathrm{LDP}$ over all valid GBs containing at least two sensitive groups.

The results are shown in Table 2. Compared with state-of-the-art GB-based methods, FairGBG consistently achieves lower $\Delta \mathrm{LDP}$ values across all datasets, reducing local unfairness to near-zero levels in most cases. In particular, on the Law dataset, FairGBG achieves a $\Delta \mathrm{LDP}$ value of 0.000, indicating that perfect local fairness is achieved within the generated GBs. On the Approval, Por, German, and Adult datasets, FairGBG reduces the average $\Delta \mathrm{LDP}$ by 97.0%, 95.0%, 97.4%, and 79.0%, respectively, compared with the runner-up results. These results indicate that the GBs generated by FairGBG maintain local demographic balance. For

the Default dataset, the $\Delta \mathrm{LDP}$ values of conventional GB-based baselines are marked as "—" because their generated GBs contain only a single sensitive group, making $\Delta \mathrm{LDP}$ incomputable. In contrast, FairGBG maintains sensitive-group diversity within GBs and achieves an average $\Delta \mathrm{LDP}$ of 0.008 on Default. These results empirically validate the effectiveness of FairGBG in constructing locally fair granular structures and further support the theoretical analysis in Theorem 3.2.

### 4.4. Comparison with Fairness-aware Methods

Since some datasets exhibit significant label imbalance, certain baseline methods degenerate into trivial classifiers (predicting a single class for all samples), rendering fairness metrics meaningless (marked as "—"). Since the compared fairness-aware methods are based on multilayer perceptrons and involve classifier training, their time cost is considerable. Therefore, we do not compare runtime and focus solely on

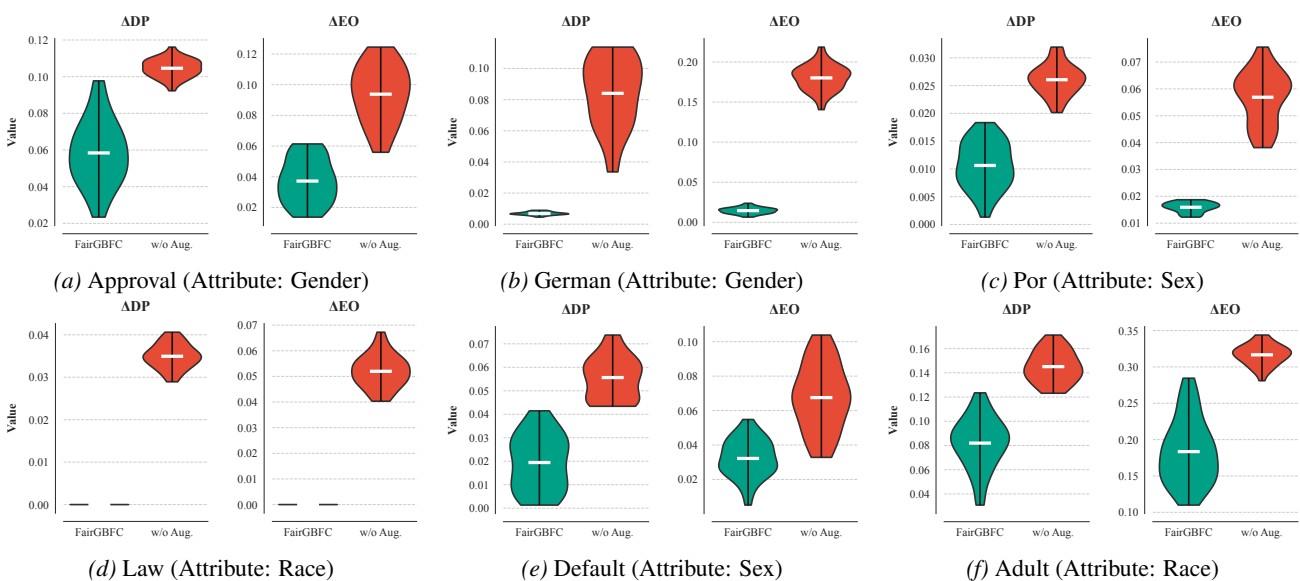

*Figure 4.* The violin plots show the distribution of ∆DP and ∆EO between FairGBFC (green) and the variant without augmentation (red), where the horizontal lines within the violins represent the mean values.

*Table 4.* Comparison of different fairness-aware methods on multi-valued sensitive attributes.

| Dataset (Task) | Method | Utility | | Fairness | |
|---|---|---|---|---|---|
| | | ACC↑ | F1↑ | DP↓ | EO↓ |
| Law (Race) | *f*-FERM | 0.948 ± 0.000 | 0.974 ± 0.000 | 0.047 ± 0.036 | 0.093 ± 0.063 |
| | Unbias | 0.949 ± 0.000 | 0.974 ± 0.000 | **0.000 ± 0.000** | **0.000 ± 0.000** |
| | DP_PP | 0.949 ± 0.001 | 0.974 ± 0.000 | 0.023 ± 0.028 | 0.054 ± 0.067 |
| | FairGBFC | **0.950 ± 0.001** | **0.975 ± 0.000** | **0.000 ± 0.000** | **0.000 ± 0.000** |
| Adult (Race) | *f*-FERM | 0.804 ± 0.003 | 0.482 ± 0.009 | 0.208 ± 0.023 | 0.569 ± 0.095 |
| | Unbias | 0.801 ± 0.006 | 0.442 ± 0.032 | 0.127 ± 0.057 | 0.434 ± 0.131 |
| | DP_PP | 0.800 ± 0.006 | 0.449 ± 0.032 | 0.130 ± 0.045 | 0.506 ± 0.137 |
| | FairGBFC | **0.805 ± 0.004** | **0.489 ± 0.028** | **0.083 ± 0.024** | **0.190 ± 0.046** |

classification and fairness metrics. Table 3 presents the comparison results between the proposed method and other fairness-aware methods on binary sensitive attributes. The results demonstrate that FairGBFC achieves the best fairness performance (△DP and △EO) across all datasets. Furthermore, with the exception of the Por dataset, FairGBFC also attains the highest classification utility. Specifically, on the Approval dataset, FairGBFC improves ACC and F1 by 0.016 and 0.015 respectively compared to the runner-up method, while reducing △DP and △EO by 16.2% and 53.2%. Similarly, on the German dataset, our method outperforms the second-best baseline with an ACC increase of 0.007, while achieving substantial fairness improvements, reducing △DP and △EO by 50.0% and 55.9%, respectively.

To further evaluate the robustness of our method, we extended the comparison to multi-valued sensitive attributes (e.g., Race), as reported in Table 4. Note that CFA and GDebias are excluded from this comparison as they are designed exclusively for binary sensitive attributes. The results indi-

cate that FairGBFC maintains superior performance in multi-group settings. On the Adult dataset, FairGBFC achieves the highest accuracy and F1 score while significantly mitigating bias, reducing △DP and △EO by approximately 34.6% and 56.2% respectively compared to the runner-up. On the Law dataset, it attains perfect fairness (0.000) alongside the highest utility. Collectively, these results confirm that FairGBFC effectively enhances both equity and utility, providing strong empirical support for the theoretical guarantees established in Theorems 3.2 and 3.3.

### 4.5. Ablation on the Impact of Sensitive Group Balance

To comprehensively evaluate the effectiveness of the Data Augmentation module within FairGBFC, we conduct an ablation study by comparing the complete framework with a variant excluding this module, denoted as "w/o Aug.". Figure 4 presents the complete ablation results across all six benchmark datasets, where the distributions of the fairness metrics ∆DP and ∆EO are compared between FairGBFC and w/o Aug.

The results consistently show that the data augmentation strategy effectively reduces fairness disparities and improves the stability of fairness performance across diverse datasets. This effect is particularly evident on the Law and Adult datasets, where the sensitive attribute is Race and contains multiple sensitive groups, posing a more challenging setting than binary sensitive attributes. On the Law dataset, the w/o Aug. variant exhibits substantial bias, whereas FairGBFC suppresses both ∆DP and ∆EO to negligible levels. Similarly, on the Adult dataset, FairGBFC achieves lower mean disparity and a more stable distribution of fair-

ness metrics. Although w/o Aug. performs worse than the complete FairGBFC, it still remains competitive with the fairness-aware baselines in Tables 3 and 4, showing that GAFC-based GB partitioning and fair inference already contribute to fairness improvement. Beyond these multi-valued sensitive attribute settings, similar improvements can also be observed on several binary-sensitive datasets. For example, on the Approval and German datasets, the w/o Aug. variant produces noticeably higher $\Delta$DP and $\Delta$EO values, while FairGBFC substantially shifts the distributions toward lower disparity regions. On the Por dataset, although the overall disparity is relatively smaller, FairGBFC still exhibits a more concentrated and lower fairness-disparity distribution than w/o Aug. These observations indicate that the Data Augmentation module contributes not only to reducing average fairness violations but also to stabilizing fairness performance across different data distributions.

Overall, these empirical results demonstrate that balancing the sensitive group distribution in the raw data is important for constructing fair GBs and achieving equitable classification, thereby empirically supporting the theoretical guarantees in Theorems 3.2 and 3.3.

## 5. Related Work

### 5.1. Group Fairness in Machine Learning

A substantial body of research has focused on enhancing group fairness in machine learning models, with most existing methods aiming to mitigate either disparate treatment or disparate impact during the prediction process. For instance, a variety of pre-processing methods (Jiang & Nachum, 2020; Chan et al., 2024; Lin et al., 2024; Shahbazi et al., 2024) and post-processing methods (Xian et al., 2023; Denis et al., 2024; Small et al., 2024; Zhang et al., 2025a) have been proposed to mitigate group bias, either by modifying the training data or by adjusting model predictions. In (Jiang & Nachum, 2020), a method is proposed that simulates the data bias generation mechanism and mitigates bias by reweighting the sample distribution. GroupDebias (Chan et al., 2024) systematically improves fairness by adaptively removing samples that contribute to increasing group disparities. DP_PP (Xian et al., 2023) builds a fair classifier by modifying prediction outputs based on a predefined scoring function to reduce disparities. In-processing methods often achieve better trade-offs between fairness and accuracy by embedding fairness constraints or debiasing mechanisms during model training (Zhao et al., 2023; Baharlouei et al., 2024; Wang et al., 2025; Halim et al., 2025; Zhang et al., 2025b). For instance, CFA (Zhao et al., 2023) reformulates the group fairness criterion, Demographic Parity, into a program-oriented constraint that is incorporated into the model optimization. Similarly, $f$-FERM (Baharlouei et al., 2024) enables fair inference even under distributional shifts.

### 5.2. Granular-Ball Computing

Inspired by the "global topology first" principle in human cognition (Chen, 1982), Granular Computing (GrC) simulates the multi-granularity information processing mechanism of the human brain (Zadeh, 1997; Wang, 2017). As a realization of GrC, Granular Ball Computing (Xia et al., 2019) characterizes data distributions using GBs rather than individual data points. By employing a coarse-to-fine splitting strategy, GBC efficiently captures both local geometry and global topology, offering a scalable and interpretable framework for data representation. GBC has demonstrated versatility across domains, including classification (Quadir & Tanveer, 2024; Xie et al., 2025), clustering (Su et al., 2025; Jia et al., 2025; Cheng et al., 2025), fuzzy set construction (Xia et al., 2024b; Lang et al., 2025), and deep learning (Liu et al., 2024a; Dai et al., 2025). Despite the efficiency of GBC in handling multi-granularity data, its potential in fairness learning remains underexplored. We propose a framework that naturally associates granule purity with group fairness by enforcing balanced sensitive group proportions during granular splitting, achieving a robust balance between fairness and accuracy without invasive data modifications.

## 6. Conclusion

In this paper, we propose FairGBFC, a novel interpretable framework for fairness-aware classification based on Granular-Ball Computing. Unlike traditional methods that often sacrifice interpretability for fairness constraints or rely on global interventions that disrupt the intrinsic data structure, FairGBFC explicitly addresses the trade-off between fairness, utility, and interpretability by constructing a multi-granularity fair space. Specifically, we generate high-purity and demographically balanced GBs through the FairGBG stage, which incorporates Neighborhood Mixup and the proposed recursive GAFC algorithm to ensure local demographic equilibrium. Building on the theoretical guarantee that maximizing purity inherently minimizes the upper bound of local unfairness, an adaptive decision mechanism is further proposed to execute fair inference by prioritizing granular regions with high structural balance. Extensive experiments on several real-world datasets demonstrate that our FairGBFC effectively improves both fairness and model performance across binary and multi-valued sensitive attribute settings, making it a promising solution for transparent and trustworthy data classification. Future work will extend FairGBFC to ultra-high-dimensional scenarios by incorporating deep latent representations or more robust similarity measures, and further integrate group fairness with individual fairness within a unified granular-ball framework to provide more comprehensive fairness guarantees.

## Acknowledgements

This work was supported by the National Key R&D Program of China under Grant 2022YFC3303200, the Guangdong Basic and Applied Basic Research Foundation under Grant No. 2026A1515011757, and the Guangdong Science and Technology Program Project under Grant No. 2025A0505020058. The corresponding author of this paper is Yuhui Deng.

## Impact Statement

This paper presents work whose goal is to advance the field of Machine Learning, specifically within the domain of algorithmic fairness and interpretable AI. The proposed framework, FairGBFC, aims to mitigate systematic biases in data-driven classification systems, which are increasingly deployed in high-stakes applications. By integrating Granular-Ball Computing with fairness constraints, our work not only improves the equity of predictive outcomes across different demographic groups but also enhances the transparency of the decision-making process. This interpretability is crucial for building public trust and accountability in automated systems. Consequently, our research has the potential to positively impact society by providing tools that help prevent discrimination against marginalized communities in automated decision pipelines.

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

*Table 5.* Summary of Main Notations

| Symbol | Description |
|---|---|
| *Dataset and Basics* | |
| $\mathcal{D}$ | The dataset containing $N$ samples, $\mathcal{D} = \{(x_i, y_i)\}_{i=1}^{N}$. |
| $N$ | Total number of samples in the dataset. |
| $x_i$ | The feature vector of the $i$-th sample, $x_i \in \mathbb{R}^d$. |
| $y_i$ | The class label of the $i$-th sample, $y_i \in \{0, 1\}$. |
| $S$ | The sensitive attribute (e.g., Race, Gender). |
| $N_S$ | Total number of sensitive groups (values) in the dataset. |
| $\mathcal{S}$ | The set of sensitive groups present in a specific context (e.g., within a GB). |
| $m$ | The count of existing sensitive groups in a subset, $m = |\mathcal{S}|$. |
| $n_{j,u}$ | Number of samples belonging to sensitive group $u$ in cluster/ball $j$. |
| *Granular-Ball Computing* | |
| $G$ | The set of generated granular balls, $G = \{gb_1, \ldots, gb_{N_{GB}}\}$. |
| $gb_j$ | The $j$-th granular ball. |
| $O_j$ | The center of granular ball $gb_j$. |
| $R_j$ | The radius of granular ball $gb_j$. |
| $n_j$ | The number of samples contained in $gb_j$. |
| $P_j$ | The purity of $gb_j$ (proportion of the majority class). |
| $L_j$ | The majority label associated with $gb_j$. |
| $\mathcal{B}_x$ | The set of granular balls that cover the query sample $x$. |
| *Fairness and Optimization* | |
| $\Delta\text{DP}$ | Global Demographic Parity difference. |
| $\Delta\text{LDP}(gb_j)$ | Label Demographic Parity difference within granular ball $gb_j$. |
| $\Delta\text{EO}$ | Equalized Odds difference. |
| $\epsilon_j$ | Structural imbalance term of $gb_j$, defined as $\epsilon_j = n_j - m_j \cdot N_{min,j}$. |
| $N_{min,j}$ | The size of the smallest sensitive group within $gb_j$. |
| $\mathcal{J}_{fair}$ | The fairness penalty term in GAFC, defined as the sum of pairwise squared count differences. |
| $\lambda$ | The fairness penalty coefficient. |
| $\lambda_{base}$ | The base hyperparameter for the fairness penalty. |
| $V_{j,u}$ | The marginal cost (gradient) of assigning a group $u$ sample to cluster $j$. |
| $\tau$ | The purity threshold for stopping the granular splitting process. |

## A. Nomenclature

Table 5 summarizes the main mathematical notations and symbols used throughout this paper.

## B. Theoretical Analysis of GAFC

In this section, we provide the theoretical proofs for the convergence and computational complexity of the proposed Group Aware Fair Clustering (GAFC) algorithm.

## B.1. Convergence Analysis of GAFC

We formally prove that the iterative Expectation-Maximization (EM) strategy employed in GAFC generates a sequence of objective function values $\{\mathcal{J}^{(t)}\}_{t=0}^{\infty}$ that is monotonically non-increasing and bounded below, thereby guaranteeing convergence to a local minimum.

**Theorem B.1** (Convergence of GAFC). *Let $\mathcal{J}^{(t)}$ denote the value of the objective function at iteration $t$. The GAFC algorithm ensures that $\mathcal{J}^{(t+1)} \leq \mathcal{J}^{(t)}$ for all $t \geq 0$.*

*Proof.* Let the objective function at iteration $t$ be defined as:

$$\mathcal{J}(\{\mathcal{C}_j\}^{(t)}, \{O_j\}^{(t)}) = \sum_{j=1}^{K} \sum_{x_i \in \mathcal{C}_j^{(t)}} \|x_i - O_j^{(t)}\|^2 + \lambda \mathcal{J}_{fair}(\{\mathcal{C}_j\}^{(t)}). \tag{10}$$

Since every term is a squared magnitude, $\mathcal{J}^{(t)} \geq 0$ is bounded below. The transition from iteration $t$ to $t+1$ consists of two distinct updates: the M-step (updating parameters) and the E-step (updating assignments). We analyze the change in the objective function for each step.

**Step 1: M-step (Parameter Update).** At the start of iteration $t$, the assignments $\{\mathcal{C}_j\}^{(t)}$ are fixed. The algorithm updates the centers to minimize the geometric loss. Let $\{O_j\}^{(t+1)}$ be the updated centers. Since $\mathcal{J}_{fair}$ depends only on counts derived from assignments and is independent of the centers, we have:

$$\{O_j\}^{(t+1)} = \arg \min_{\{O_j\}} \sum_{j=1}^{K} \sum_{x_i \in \mathcal{C}_j^{(t)}} \|x_i - O_j\|^2. \tag{11}$$

The solution is the sample mean of the current clusters. By definition of the minimum, the objective function value with the new centers must be less than or equal to the value with the old centers:

$$\mathcal{J}(\{\mathcal{C}_j\}^{(t)}, \{O_j\}^{(t+1)}) \leq \mathcal{J}(\{\mathcal{C}_j\}^{(t)}, \{O_j\}^{(t)}). \tag{12}$$

Simultaneously, the demographic potentials $V_{j,u}^{(t+1)}$ are updated based on the fixed assignments $\{\mathcal{C}_j\}^{(t)}$ to prepare for the next step.

**Step 2: E-step (Assignment Update).** With the updated centers $\{O_j\}^{(t+1)}$ and potentials $V_{j,u}^{(t+1)}$ fixed, the algorithm updates the assignments to $\{\mathcal{C}_j\}^{(t+1)}$. Each sample $x_i$ is reassigned to the cluster $j^*$ that minimizes the linearized cost:

$$j^* = \arg \min_{j} \left( \|x_i - O_j^{(t+1)}\|^2 + \lambda V_{j,s_i}^{(t+1)} \right). \tag{13}$$

This assignment rule corresponds to a block coordinate descent step. Specifically, $V_{j,s_i}^{(t+1)}$ represents the gradient of the fairness term with respect to the cluster counts. Moving a sample to a cluster with a lower potential reduces the pairwise fairness penalty $\mathcal{J}_{fair}$. Thus, the re-assignment ensures that the combined cost for each sample decreases. Aggregating over all samples, the total objective function decreases (or remains constant):

$$\mathcal{J}(\{\mathcal{C}_j\}^{(t+1)}, \{O_j\}^{(t+1)}) \leq \mathcal{J}(\{\mathcal{C}_j\}^{(t)}, \{O_j\}^{(t+1)}). \tag{14}$$

Combining inequalities (12) and (14), we obtain the monotonic property for the full iteration:

$$\mathcal{J}^{(t+1)} \leq \mathcal{J}(\{\mathcal{C}_j\}^{(t)}, \{O_j\}^{(t+1)}) \leq \mathcal{J}^{(t)}. \tag{15}$$

Since the sequence $\{\mathcal{J}^{(t)}\}$ is monotonically non-increasing and bounded below by 0, it must converge to a local minimum according to the Monotone Convergence Theorem. $\square$

## B.2. Time Complexity Analysis

We analyze the computational complexity of GAFC per iteration. Let $N$ be the number of samples, $d$ be the feature dimension, $K$ be the number of clusters, and $m$ be the number of sensitive groups present in the dataset.

**Theorem B.2** (Time Complexity). *The time complexity of GAFC for one iteration is $\mathcal{O}(N \cdot K \cdot d)$.*

*Proof.* The complexity is determined by the operations in the M-step and E-step.

**1. M-step (Updates):**

- **Updating Centers:** Calculating the mean vector for $K$ clusters requires summing vectors for all $N$ samples. Cost: $\mathcal{O}(N \cdot d)$.

- **Updating Potentials:** Calculating the group counts $n_{j,u}$ takes $\mathcal{O}(N)$. Computing potentials $V_{j,u}$ for $K$ clusters and $m$ groups takes $\mathcal{O}(K \cdot m)$. Since $m \ll N$, this is negligible.

**2. E-step (Assignment):** We iterate through all $N$ samples to determine the optimal cluster assignment.

- **Distance Calculation:** Euclidean distance $\|x_i - O_j\|^2$ takes $\mathcal{O}(d)$.

- **Potential Lookup:** Accessing the pre-computed value $V_{j,s_i}$ takes $\mathcal{O}(1)$.

- **Selection:** For each sample, we evaluate $K$ clusters. Total cost: $\mathcal{O}(N \cdot K \cdot d)$.

**Total Complexity:** Summing these components, the total complexity per iteration is dominated by the distance calculations:

$$T_{total} \approx \mathcal{O}(N \cdot K \cdot d). \tag{16}$$

Crucially, the fairness computation is efficiently decoupled via the demographic potentials, ensuring that the complexity does not scale quadratically with the number of groups or samples. This linear scalability makes GAFC highly efficient for recursive applications. $\square$

## C. Algorithm Details

In this section, we provide the detailed pseudocode for the underlying optimization strategy. Algorithm 1 details the Group Aware Fair Clustering (GAFC) algorithm, which serves as the core splitting engine, and Algorithm 2 presents the overall Fair GB Generation (FairGBG) process that utilizes it.

## D. Proof of Theorem 3.2

We provide the rigorous derivation for the upper bound of the Multi-Group Label Demographic Parity (LDP) within a specific GB $gb_j$.

*Proof.* Let $gb_j$ be a GB containing $n_j$ samples. Let $\mathcal{S}_j$ be the set of sensitive groups in $gb_j$, with cardinality $m_j = |\mathcal{S}_j|$. For each group $u \in \mathcal{S}_j$, let $n_{j,u}$ denote its sample size. We define $N_{min,j} = \min_{u \in \mathcal{S}_j} n_{j,u}$. The total sample size can be decomposed as:

$$n_j = \sum_{u \in \mathcal{S}_j} n_{j,u} = m_j \cdot N_{min,j} + \epsilon_j, \tag{17}$$

where $\epsilon_j \geq 0$ represents the total excess samples contributing to structural imbalance. Let $L_j$ be the majority label of $gb_j$. By the definition of purity $P_j$, the number of samples with the minority label (i.e., classification errors if we assign $L_j$ to the entire ball) is:

$$N_{err} = n_j(1 - P_j). \tag{18}$$

The LDP metric is defined as the maximum pairwise difference in positive rates:

$$\Delta\text{LDP}(gb_j) = \max_{u,v \in \mathcal{S}_j} |P(Y = 1 \mid S = u) - P(Y = 1 \mid S = v)|. \tag{19}$$

---

**Algorithm 1** Group Aware Fair Clustering (GAFC)

---

1: **Input:** Dataset $\mathcal{D}$ (subset of samples), number of clusters $K$, fairness penalty $\lambda$, max iterations $T_{max}$.
2: **Output:** Disjoint clusters $\{\mathcal{C}_1, \ldots, \mathcal{C}_K\}$.
3: **Initialization:**
4: Initialize centers $\{O_j^{(0)}\}_{j=1}^K$ (e.g., via K-means++ or random selection).

5: Initialize cluster assignments $\{\mathcal{C}_j^{(0)}\}$ arbitrarily.
6: Determine the set of present sensitive groups $\mathcal{S}$ and count $m = |\mathcal{S}|$.
7: $t \leftarrow 0$
8: **while** not converged **and** $t < T_{max}$ **do**
9:     *// M-step: Update Parameters for iteration $t + 1$*
10:     **for** $j = 1$ **to** $K$ **do**
11:         $O_j^{(t+1)} \leftarrow \frac{1}{|\mathcal{C}_j^{(t)}|} \sum_{x_i \in \mathcal{C}_j^{(t)}} x_i$
12:         **for** each group $u \in \mathcal{S}$ **do**
13:             Calculate count $n_{j,u}^{(t)} \leftarrow \sum_{x_i \in \mathcal{C}_j^{(t)}} \mathbb{I}(s_i = u)$
14:             $V_{j,u}^{(t+1)} \leftarrow m \cdot n_{j,u}^{(t)} - |\mathcal{C}_j^{(t)}|$
15:         **end for**
16:     **end for**
17:     *//E-step: Batch Assignment for iteration $t + 1$*
18:     Clear clusters: $\mathcal{C}_j^{(t+1)} \leftarrow \emptyset$ for all $j$.
19:     **for** each sample $x_i \in \mathcal{D}$ with group $s_i$ **do**
20:         Find best cluster index $j^*$:

$$j^* \leftarrow \arg \min_{j \in \{1,\ldots,K\}} \left( \|x_i - O_j^{(t+1)}\|^2 + \lambda \cdot V_{j,s_i}^{(t+1)} \right)$$

21:         Assign sample: $\mathcal{C}_{j^*}^{(t+1)} \leftarrow \mathcal{C}_{j^*}^{(t+1)} \cup \{x_i\}$
22:     **end for**
23:     *// Check Convergence*
24:     **if** $\{\mathcal{C}_j^{(t+1)}\} = \{\mathcal{C}_j^{(t)}\}$ **then**
25:         **break**
26:     **end if**
27:     $t \leftarrow t + 1$
28: **end while**
29: **return** $\{\mathcal{C}_1^{(t)}, \ldots, \mathcal{C}_K^{(t)}\}$

---

We analyze the bound by considering the distribution of minority label samples across groups. Let $e_u$ denote the number of minority label samples within group $u$. Note that $\sum_{u \in \mathcal{S}_j} e_u = N_{err}$.

**Case 1: The majority label is Positive ($L_j = 1$).** The positive rate for group $u$ is $P(Y = 1 \mid S = u) = \frac{n_{j,u} - e_u}{n_{j,u}} = 1 - \frac{e_u}{n_{j,u}}$. The pairwise difference becomes:

$$\left| \left(1 - \frac{e_u}{n_{j,u}}\right) - \left(1 - \frac{e_v}{n_{j,v}}\right) \right| = \left| \frac{e_v}{n_{j,v}} - \frac{e_u}{n_{j,u}} \right|. \tag{20}$$

Since $n_{j,k} \geq N_{min,j}$ for all $k$, we have $\frac{e_k}{n_{j,k}} \leq \frac{e_k}{N_{min,j}}$. Thus:

$$\left| \frac{e_v}{n_{j,v}} - \frac{e_u}{n_{j,u}} \right| \leq \max \left( \frac{e_v}{n_{j,v}}, \frac{e_u}{n_{j,u}} \right) \leq \frac{\max(e_v, e_u)}{N_{min,j}}. \tag{21}$$

Since the errors in a single group cannot exceed the total errors in the ball (i.e., $\max(e_v, e_u) \leq N_{err}$), we obtain:

$$\Delta \text{LDP}(gb_j) \leq \frac{N_{err}}{N_{min,j}}. \tag{22}$$

---

**Algorithm 2** Fair Granular Ball Generation (FairGBG)

---

1: **Input:** Augmented dataset $\mathcal{D}_{aug}$, Purity threshold $\tau$, Total sensitive groups $N_S$, Base penalty parameter $\lambda_{base}$.
2: **Output:** A set of fair GBs $G_{fair}$.
3: **Initialization:**
4: $G_{candidate} \leftarrow \emptyset$
5: Create root GB $gb_{root}$ containing all samples in $\mathcal{D}_{aug}$.
6: Initialize queue $\mathcal{Q} \leftarrow \{gb_{root}\}$
7: **while** $\mathcal{Q}$ is not empty **do**
8:     Pop a GB $gb_j$ from $\mathcal{Q}$.
9:     Calculate purity $P_j$ and sample size $n_j$.
10:    **if** $P_j \geq \tau$ **then**
11:        *// High quality: Temporarily retain*
12:        $G_{candidate} \leftarrow G_{candidate} \cup \{gb_j\}$
13:    **else if** $n_j \geq N_S$ **then**
14:        *// Low quality but splittable: Apply adaptive split*
15:        *// 1. Compute Geometric Magnitude*
16:        $\bar{d}_j \leftarrow \frac{1}{n_j} \sum_{x \in gb_j} \|x - O_j\|$, where $O_j$ is the center of $gb_j$.
17:        *// 2. Compute Demographic Imbalance Degree*
18:        Identify active groups $\mathcal{S}_j$ in $gb_j$ and count $m_j = |\mathcal{S}_j|$.
19:        $\mathcal{I}_j \leftarrow \sum_{u \in \mathcal{S}_j} \left( \frac{n_{j,u}}{n_j} - \frac{1}{m_j} \right)^2$.
20:        *// 3. Determine Adaptive Penalty*
21:        $\lambda_j \leftarrow \lambda_{base} \cdot (1 + \mathcal{I}_j) \cdot \bar{d}_j$
22:        Partition samples in $gb_j$ into $\{\mathcal{C}_1, \mathcal{C}_2\}$ using **GAFC** (Algorithm 1) with $K = 2$ and penalty $\lambda_j$.
23:        **for** $k \in \{1, 2\}$ **do**
24:            Construct child ball $gb_{child\_k}$ from cluster $\mathcal{C}_k$.
25:            Push $gb_{child\_k}$ into $\mathcal{Q}$.
26:        **end for**
27:    **else**
28:        *// Pruning Mechanism: Low quality and unsplittable*
29:        Discard $gb_j$.
30:    **end if**
31: **end while**
32: **Final Filtration Phase:**
33: $G_{fair} \leftarrow \{gb \in G_{candidate} \mid gb.\text{purity} \geq \tau \text{ AND } gb.\text{size} \geq N_S\}$
34: **return** $G_{fair}$

---

**Case 2: The majority label is Negative** ($L_j = 0$)**.** The positive rate is $P(Y = 1 \mid S = u) = \frac{e_u}{n_{j,u}}$. The difference is simply $\left| \frac{e_u}{n_{j,u}} - \frac{e_v}{n_{j,v}} \right|$, which follows the exact same bounding logic as Case 1.

**Synthesis:** Combining both cases, we have the general bound:

$$\Delta \text{LDP}(gb_j) \leq \frac{n_j(1 - P_j)}{N_{min,j}}. \tag{23}$$

Substituting $n_j = m_j N_{min,j} + \epsilon_j$ into the inequality:

$$\Delta \text{LDP}(gb_j) \leq \frac{m_j N_{min,j} + \epsilon_j}{N_{min,j}} (1 - P_j) = \left( m_j + \frac{\epsilon_j}{N_{min,j}} \right)(1 - P_j). \tag{24}$$

Expanding the term:

$$\Delta \text{LDP}(gb_j) \leq m_j(1 - P_j) + \frac{\epsilon_j}{N_{min,j}}(1 - P_j). \tag{25}$$

In binary classification, the purity of the majority class satisfies $P_j \geq 0.5$, which implies $(1 - P_j) \leq 0.5$. Applying this

inequality specifically to the second term (the structural imbalance term):

$$\Delta\text{LDP}(gb_j) \le m_j(1 - P_j) + \frac{\epsilon_j}{2N_{min,j}}. \tag{26}$$

This completes the proof. □

## E. Proof of Theorem 3.3

*Proof.* **1. Notation and Preliminaries**

Let $N$ be the total number of samples in the dataset and $N_S$ be the number of sensitive groups. By the assumption of global balance, the size of each sensitive group $u \in \{1, \ldots, N_S\}$ is equal:

$$N_u = \frac{N}{N_S}, \quad \forall u. \tag{27}$$

The feature space is partitioned into $N_{GB}$ disjoint decision regions $\{\mathcal{R}_j\}_{j=1}^{N_{GB}}$, where each region corresponds to a GB $gb_j$ and is assigned a uniform label $L_j \in \{0, 1\}$. Let $n_{j,u}$ denote the number of samples from group $u$ falling into region $\mathcal{R}_j$. Note that if group $u$ is not represented in $gb_j$, then $n_{j,u} = 0$.

**2. Decomposition of Global Demographic Parity**

The Demographic Parity (DP) is defined as the maximum difference in positive prediction rates between any two groups $u$ and $v$:

$$\Delta\text{DP} = \max_{u,v} \left| P(\hat{Y} = 1 \mid S = u) - P(\hat{Y} = 1 \mid S = v) \right|. \tag{28}$$

The positive prediction rate for group $u$ can be decomposed as the sum of contributions from each decision region where the assigned label is positive ($L_j = 1$):

$$P(\hat{Y} = 1 \mid S = u) = \frac{1}{N_u} \sum_{j=1}^{N_{GB}} n_{j,u} \cdot \mathbb{I}(L_j = 1). \tag{29}$$

Substituting $N_u = N/N_S$, the pairwise difference becomes:

$$\left| P(\hat{Y} = 1 \mid S = u) - P(\hat{Y} = 1 \mid S = v) \right| = \frac{N_S}{N} \left| \sum_{j:L_j=1} (n_{j,u} - n_{j,v}) \right|. \tag{30}$$

By the triangle inequality, we can move the absolute value inside the summation to obtain an upper bound:

$$\Delta\text{DP} \le \frac{N_S}{N} \sum_{j=1}^{N_{GB}} |n_{j,u} - n_{j,v}|. \tag{31}$$

This summation runs over all $j$, which constitutes a valid upper bound because terms where $L_j = 0$ contribute 0, and adding non-negative absolute differences for all regions covers the worst-case assignment.

**3. Bounding Local Group Differences**

We now bound the term $|n_{j,u} - n_{j,v}|$ for a specific ball $gb_j$ based on its structural properties. Recall the definitions for ball $gb_j$:

- $m_j$: The number of sensitive groups present in $gb_j$.

- $N_{min,j} = \min_{k \in \mathcal{S}_j} n_{j,k}$: The base count of the smallest present group.

- $\epsilon_j = \sum_{k \in \mathcal{S}_j} (n_{j,k} - N_{min,j})$: The structural imbalance (excess samples).

For any group $k$, the count $n_{j,k}$ can be expressed as:

$$n_{j,k} = \begin{cases} N_{min,j} + \delta_{j,k} & \text{if group } k \text{ is present,} \\ 0 & \text{if group } k \text{ is missing,} \end{cases} \tag{32}$$

where $\delta_{j,k} \geq 0$ is the excess count, and $\sum \delta_{j,k} = \epsilon_j$.

Consider the worst-case difference $|n_{j,u} - n_{j,v}|$. The maximum discrepancy arises when we compare a group with the highest count against a missing group.

- If $u$ is present and $v$ is missing: $|n_{j,u} - 0| = N_{min,j} + \delta_{j,u}$. Since $\delta_{j,u} \leq \epsilon_j$, this is bounded by $N_{min,j} + \epsilon_j$.

To derive a bound that accounts for the contribution of missing groups to the global calculation, we observe that the term $(N_S - m_j)N_{min,j}$ represents the total missing mass penalty for ball $j$. Mathematically, the difference $|n_{j,u} - n_{j,v}|$ across the dataset is dominated by the structural imbalance $\epsilon_j$ plus the penalty for missing groups:

$$|n_{j,u} - n_{j,v}| \leq \epsilon_j + (N_S - m_j)N_{min,j}. \tag{33}$$

The term $\epsilon_j$ bounds the difference arising from the excess samples among present groups, while $(N_S - m_j)N_{min,j}$ accounts for the structural gap created by the empty slots of the missing groups.

### 4. Final Assembly

Substituting this local bound back into Eq. (31):

$$\Delta\text{DP} \leq \frac{N_S}{N} \sum_{j=1}^{N_{GB}} \left( \epsilon_j + (N_S - m_j)N_{min,j} \right). \tag{34}$$

In the ideal case where all sensitive groups are present in every GB (i.e., full coverage where $m_j = N_S$), the missing group penalty term becomes zero. The bound strictly simplifies to:

$$\Delta\text{DP} \leq \frac{N_S}{N} \sum_{j=1}^{N_{GB}} \epsilon_j. \tag{35}$$

This completes the proof. $\qquad\square$

## F. Dataset Details

### F.1. Dataset Descriptions

Table 6 summarizes the detailed statistics of the six benchmark datasets used in our experiments, including the domain, sample size, positive ratio, and the specific distribution of sensitive groups. **Approval Dataset (Quinlan, 1987):** The Approval (Credit Approval) dataset comprises 690 samples and 15 attributes. It is typically used to predict whether a credit card application will be approved based on personal information. In our experiments, we select *Gender* as the sensitive attribute.

**German Dataset (Dua & Graff, 2017):** The German Credit dataset contains 1,000 instances representing individuals classified as good or bad credit risks. It includes 20 attributes covering financial and demographic details. We designate *Gender* as the sensitive attribute for fairness evaluation.

**Por Dataset (Silva, 2008):** The Student Performance (Por) dataset contains data on student achievement in secondary education of two Portuguese schools. It consists of 649 instances with 33 features including student grades, demographic, social and school-related features. The task is to predict the final grade, and we treat *Sex* as the sensitive attribute.

**Law Dataset (Wightman, 1998):** The Law dataset, provided by the Law School Admission Council, contains detailed information on 21,790 students from 163 law schools in the United States. We select seven features relevant to predicting whether a law school applicant is likely to pass the bar exam. In our experiments, we utilize both *Gender* and *Race* as sensitive attributes to evaluate fairness across different dimensions.

*Table 6.* Dataset statistics. The "Group Distribution" column presents the Sample Ratio and (Positive Ratio) for each group within the sensitive attribute.

| Dataset | Sens. Attr. | Overall Pos. Ratio | Group Distribution: Sample Ratio (Positive Ratio) |
|---------|-------------|--------------------|--------------------------------------------------|
| *Approval* | Gender | 44.5% | 30.43% (46.67%), 69.57% (43.54%) |
| *German* | Gender | 70.0% | 69.00% (72.32%), 31.00% (64.84%) |
| *Por* | Sex | 15.4% | 59.01% (13.05%), 40.99% (18.80%) |
| *Default* | Sex | 77.9% | 39.63% (75.83%), 60.37% (79.22%) |
| *Adult* | Sex | 24.9% | 32.43% (11.37%), 67.57% (31.38%) |
| | Race | | 0.95% (11.89%), 2.97% (27.71%), 9.34% (12.99%), 0.77% (9.09%), 85.98% (26.37%) |
| *Law* | Gender | 94.9% | 43.59% (94.18%), 56.41% (95.41%) |
| | Race | | 3.85% (92.40%), 5.89% (77.44%), 4.52% (87.70%), 1.82% (90.26%), 83.92% (96.70%) |

**Default Dataset (Yeh, 2009):** The Default dataset originates from a major bank in Taiwan. It contains records of 30,000 credit card holders and includes 23 features covering demographic information, credit limits, billing details, historical repayment behavior, and default status. This dataset is used primarily to predict whether a user will default on their credit card payment. We treat *Sex* as the sensitive attribute.

**Adult Dataset (Dua & Graff, 2017):** The Adult (Census Income) dataset is extracted from the 1994 U.S. Census database. It contains 48,842 instances with 14 attributes such as age, work class, education, and occupation. The classification task is to predict whether an individual's annual income exceeds $50K. We conduct experiments using both *Sex* and *Race* as sensitive attributes.

### F.2. Data Preprocessing

We first perform data deduplication and apply Min-Max normalization to scale all numerical features to the range $[0, 1]$. For datasets with larger sample sizes (e.g., Law, Default, and Adult), samples with missing values are removed to ensure data quality. In contrast, for smaller datasets (e.g., Approval, German, Por), missing values are handled using mean imputation for continuous features and mode imputation for categorical features.

For the Law dataset, we specifically select eight features following standard protocols: LSAT score (lsat), undergraduate GPA (ugpa), first-year graduate GPA (zfygpa), cumulative GPA (zgpa), full-time status (fulltime), gender (gender), race (race1), and the target feature indicating whether the bar exam was passed (pass_bar). For the Approval and Default datasets, we utilize all available features. For the Por and Math datasets, we adopt the same feature settings as described in (Zhao et al., 2023). For the Adult dataset, we retain only ten attributes: workclass, education, marital-status, occupation, relationship, race, sex, native-country, income, and age. Categorical features across all datasets are transformed into numerical representations using one-hot encoding.

## G. Implementation Details and Hyperparameter Settings

### G.1. Hardware and Software Environment

All methods in this study are implemented in Python 3.10. The experiments are conducted on a standard computing server equipped with an Intel(R) Xeon(R) Silver 4216 CPU running at 2.10 GHz and 128 GB of RAM. The operating system is the Community Enterprise Operating System (CentOS) 7.0, powered by a 4.18.0 kernel.

### G.2. Hyperparameter Configuration

To ensure a fair and rigorous comparison, all baseline methods are implemented using the recommended parameter settings and architectures specified in their respective original papers. For our proposed FairGBFC method, there are two key

hyperparameters requiring configuration: the base fairness weight ($\lambda_{base}$) and the purity threshold ($\tau$).

- **Purity Threshold** ($\tau$)**:** This parameter controls the stopping criterion for the granular splitting process. In our experiments, $\tau$ is selected via grid search within the range of $[0.90, 0.99]$. A higher $\tau$ yields finer-grained balls with higher purity, while a slightly lower $\tau$ encourages larger, more generalized granular structures.

- **Base Fairness Weight** ($\lambda_{base}$)**:** This parameter determines the strength of the fairness penalty during the Group Aware Fair Clustering (GAFC) step. Its optimal range depends on the specific characteristics of the dataset. Specifically, for datasets requiring subtle fairness adjustments, $\lambda_{base}$ is selected from $[0.1, 1]$, while for datasets with significant demographic imbalances, it is tuned within the range of $[1, 100]$.

