# OpenReview forum: "FairGB: A Fair Granular-Ball Generation Method for Data Classification"
_ICML.cc/2026/Conference — ICML 2026 regular_

### Official Review · Reviewer_4yfC · 2026-03-09

**Soundness:** 4
**Presentation:** 3
**Significance:** 3
**Originality:** 4
**Overall Recommendation:** 5
**Confidence:** 4

**Summary:**

The authors tackle the problem of fair clustering and classification through the FairGBC and FairGBFC frameworks, leveraging the use of granular balls along with data augmentation in order to achieve a fair clustering algorithm which can be extended to handle statistical classification by preserving label proportion across demographic groups during the data augmentation. The fairness properties of the algorithm are established through a series of bounds that follow from the structural properties of the construction of the granular balls, allowing to achieve fair clustering and classification through local-to-global bounds. These properties are empirically assessed on six real-world datasets against other granular-ball based and fairness-aware algorithms in the context of fair classification.

**Compliance With Llm Reviewing Policy:**

Affirmed.

**Final Justification:**

In my original review I shared many reasons I believe the work is both original and technically sound within the realm of fair Machine Learning, offering a joint solution to the problems of fair clustering and classification through an approach based in Granular Balls. This formalism allowed the authors to paint a compelling picture on the strengths of their model, both by providing a thorough theoretical exposition with strong results validating their intuition and a solid empirical assesment of the capabilities of their models, comparing their approach with many other baselines in common fairness datasets both in the realm of clustering and classification. Overall, the paper presented its findings in a concise and clear manner.

During this review I raised three concerns revolving around the empirical assesment, two of which required additional experiments or evaluations. This is why at the time I felt it was fair to score the paper with a **weak accept (4)**. However, the authors responded to all of my concerns and showed the results of the additional experiments required by my inquiries, thus shedding light into what I believe was the weakest aspect of their work. The inclusion of these additional evaluations in the revised paper completes the experimental setting and gives answer to basic questions previously unanswered. This is why I am rising the previous score up to an **accept (5)**.

Finally, I want to thank the authors to take the time to thoroughly respond to all of my concerns.

**Key Questions For Authors:**

1. Could the presented results be explained and reproduced simply with the data augmentation using a different algorithm?
2. How does the FairGBFC method compare against established methods in the literature of fair machine learning?
3. Do the authors intend to address the close-to-ideal results for the Law dataset?

**Limitations:**

yes

**Strengths And Weaknesses:**

Strengths:

The authors propose an original application of granular balls to achieve a principled, intuitive framework addressing the problem of fair clustering and classification. Both the overall idea and the mathematical formalization are explained in a clear, concise manner in the main text. In particular, the properties of the framework arise from the structural properties of the construction of the granular balls, which are ensured through a previous step consisting of data augmentation, and provide bounds on the fairness of the resulting classifier through a local-to-global argument.

These ideas are put to the test in a wide array of empirical experiments assessing the capabilities of the model in diverse settings, establishing the performance of the proposed approach against many different baselines in the realms of both clustering and classification. Moreover, the authors carefully report the execution time for both their methods and the baselines, where their approaches dominate. All of these factors show that the granular ball framework as formulated by the authors is sound both theoretically and empirically, with great care put into the model performance.

All in all, the authors propose an intuitive, efficient approach to the problems of fair clustering and classification through an original application of granular balls, establishing their method in the landscape of fair machine learning.

Weaknesses:

The paper, while solid, admits polishing. In particular, most of the criticism I find with the methodology lies in the experimental Section.

First, the proposed approach is built on two pillars: granular balls and data augmentation. Section 4.4 highlights an ablation study demonstrating the importance of the data augmentation procedure. However, the role of the granular ball itself is not put to the test, thus creating the question on whether or not a simple model like logistic regression could achieve similar results with the data augmentation pre-processing step.

Second, the comparison against fairness-aware methods, while satisfactory, does not take into account more established methods in the fairness literature. Some key references that come to mind include reweighting (Kamiran and Calders), adversarial debiasing (Zhang et al) or the equal odds processor (Hardt et al), which are readily available in the aif360 library.

Finally, the results for the Law dataset seem close to ideal, with accuracy close to one and unfairness metrics close to zero. Of course, this could be a result of the nature of the data itself. However, it could also hint to an issue during data processing like label leakage. In any case, this point should be addressed in order to calm the doubts this might arise in more skeptic readers.

---

> ### Author Rebuttal · Authors · 2026-03-31
>
> We thank the reviewer for recognizing the originality and mathematical clarity of our Granular-Ball (GB) framework. Below, we address your questions regarding the experimental methodology in detail:
>
> **Q1: The role of the GB itself is not put to the test, thus creating the question on whether or not a simple model like logistic regression could achieve similar results with the data augmentation pre-processing step.**
>
> **Response**: To isolate and verify the role of the GB framework, we applied our exact data augmentation pre-processing step to standard Logistic Regression (LR) and Multi-Layer Perceptron (MLP) models. The mean results are shown below:
>
> | Dataset | Method | Acc | F1 | $\Delta\text{DP}$ | $\Delta\text{EO}$ |
> | :--- | :--- | :--- | :--- | :--- | :--- |
> | Approval (Gender) | LR | 0.841 | 0.832 | 0.096 | 0.088 |
> | | MLP | 0.836 | 0.813 | 0.123 | 0.174 |
> | Law (Gender) | LR | 0.948 | 0.974 | 0.007 | 0.054 |
> | | MLP | 0.946 | 0.974 | 0.009 | 0.059 |
> | Por (Sex) | LR | 0.854 | 0.448 | 0.064 | 0.239 |
> | | MLP | 0.837 | 0.000 | N/A | N/A |
> | German (Gender) | LR | 0.714 | 0.821 | 0.078 | 0.149 |
> | | MLP | 0.702 | 0.806 | 0.142 | 0.198 |
> | Adult (Sex) | LR | 0.763 | 0.237 | 0.108 | 0.182 |
> | | MLP | 0.808 | 0.560 | 0.226 | 0.176 |
> | Default (Sex) | LR | 0.801 | 0.879 | 0.030 | 0.047 |
> | | MLP | 0.810 | 0.889 | 0.023 | 0.038 |
>
> Compared to FairGBFC (**Table 2** in the manuscript), data augmentation alone fails to achieve a strong fairness-utility trade-off. Simple models severely struggle to leverage the augmented data effectively, leading to significant utility degradation (e.g., LR F1=0.237 on Adult; MLP F1=0.000 on Por). Furthermore, without the structural constraints of GBs, simple models fail to strictly bound unfairness; for instance, on German, $\Delta\text{EO}$ for LR/MLP is ~10x worse than that of FairGBFC. Thus, while augmentation balances sensitive group distributions without altering positive rates, the GB structure is essential for decision boundaries that ensure both fairness and accuracy.
>
> **Q2: Comparison Against Established Fairness Baselines. How does the FairGBFC method compare against established methods in the literature of fair machine learning?**
>
> **Response:** As suggested, we have evaluated the three fairness methods from the AIF360 library (Reweighting (Rw) (Kamiran and Calders, 2012), Equal Odds Processor (EqOdds) (Hardt et al., 2016), and Adversarial Debiasing (Adv) (Zhang et al., 2018)) using the exact same experimental setup. The results are summarized below:
>
> | Dataset|Method|Acc| F1 | $\Delta\text{DP}$ | $\Delta\text{EO}$ |
> | :--- | :--- | :--- | :--- | :--- | :--- |
> | Approval (Gender)| EqOdds | 0.825 | 0.796 | 0.068 | 0.082 |
> | | Adv | 0.855 | 0.846 | 0.071 | 0.087 |
> | Law (Gender)| Rw | 0.949 | 0.974 | 0.002 | 0.019 |
> | | EqOdds | 0.949 | 0.974 | 0.001 | 0.030 |
> | | Adv | 0.949 | 0.974 | 0.002 | 0.025 |
> |Por(Sex)| Rw | 0.846 | 0.000 | N/A | N/A |
> | | EqOdds | 0.846 | 0.000 | N/A | N/A |
> | | Adv | 0.868 | 0.505 | 0.084 | 0.349 |
> | German (Gender)  | Rw | 0.703 | 0.821 | 0.016 | 0.053 |
> | | EqOdds | 0.622 | 0.655 | 0.026 | 0.059 |
> | | Adv | 0.716 | 0.816 | 0.024 | 0.096 |
> | Adult (Sex) | Rw| 0.802 | 0.509 | 0.092 | 0.085 |
> | | EqOdds | 0.777 | 0.457 | 0.096 | 0.088 |
> | | Adv | 0.748 | 0.084 | 0.100 | 0.099 |
> | Default (Sex) | Rw | 0.819 | 0.891 | 0.031 | 0.041 |
> | | EqOdds | 0.813 | 0.887 | 0.021 | 0.034 |
> | | Adv | 0.810 | 0.889 | 0.020 | 0.039 |
>
> Compared to FairGBFC (**Table 2**), these baselines show severe instability and inferior trade-offs. Rw and EqOdds suffer total utility collapse on the Por dataset (F1=0.000). Meanwhile, Adv collapses on Adult (F1=0.084) and yields massive $\Delta\text{EO}$ errors (0.349) on Por. Conversely, FairGBFC robustly avoids collapse, maintaining high utility while effectively mitigating bias.
>
> **Q3: Addressing the Results on the Law Dataset. Do the authors intend to address the close-to-ideal results for the Law dataset?**
>
> **Response:** We confirm that our preprocessing steps are standard and completely prevent label leakage: feature normalization is fitted exclusively on the training set before being applied to the test set. The near-ideal metrics stem from the Law dataset's inherent skew (Appendix Table 5 shows >94% positive rates across groups). Our augmentation preserves this imbalance where classifiers easily achieve high accuracy by defaulting to the positive class. Indeed, baselines achieve ~0.94 accuracy on Law-Gender, and CFA/Unbias reach 0.000 fairness errors, confirming this reflects the true data manifold. Still, this dataset is a rigorous stress test; strictly aligning prediction rates across 5 Law-Race groups remains challenging. Here, $f$-FERM and DP_PP leave residual $\Delta\text{DP}$ errors of 0.047 and 0.023, and GDebias retains high $\Delta\text{EO}$ on Law-Gender. Conversely, FairGBFC stably achieves 0.000 across metrics. By utilizing fair GBs, our method handles extreme distributions with exceptional stability.

---

> > ### Author Rebuttal · Reviewer_4yfC · 2026-04-02
> >
> > I want to thank the authors for considering all my criticism and addressing all my concerns, specially considering they required the addition of new experiments on a tight deadline. I believe the inclusion of these experiments, specially the ablation on the data augmentation step, allows the paper to paint a more complete picture of the proposed approach, which was already compelling both empirically and theoretically.
> >
> > I believe the reasons listed above merit increasing my original score (4) up to a 5.

---

> > > ### Author Response · Authors · 2026-04-03
> > >
> > > We sincerely thank you for your time, your constructive feedback throughout the review process, and for acknowledging our rebuttal efforts. We are very glad that the new experiments, especially the ablation study, have successfully addressed your concerns and helped present a more complete picture of our work. We will make sure to incorporate all these new results prominently in the final version of the paper. Thank you again for championing our work and for adjusting the score!

---

### Official Review · Reviewer_7fBk · 2026-03-10

**Soundness:** 3
**Presentation:** 3
**Significance:** 3
**Originality:** 2
**Overall Recommendation:** 4
**Confidence:** 4

**Summary:**

This paper proposes FairGBFC, a fair algorithm based on granular-ball computing.
The main idea is to first construct fair granular balls through data augmentation (using mixup technique) and fair clustering, and then perform classification.
Classification step consists of adaptively using the label purity and sensitive group balance of those balls.
The paper provides theoretical analysis showing that increasing the purity of granular balls and balancing the group ratios can reduce an upper bound on unfairness, and also presents experiments on several real-world datasets.

**Compliance With Llm Reviewing Policy:**

Affirmed.

**Final Justification:**

I think this paper is good overall, for example, theoretical results provide an interesting insight.
The authors further provided detailed responses to the reviews, hence, I will maintain the positive rating.

**Key Questions For Authors:**

- Please see "Weaknesses" described above.
- The data augmentation step builds “globally equalized” groups, i.e., $N_1 = N_{N_S} = N / N_S$. However, does this step distort the true/observed ratio of each group (i.e., $P(S = v)$)? Furthermore, the upper bound of $\Delta DP$ depends on $m_j$ which is expected to be close to $N_s.$ Can the authors explain behind the gap between (1) perturbing the true ratio of each group and (2) the theoretical results? More specifically, how can the proposed method still achieve fairness when the distribution of the augmented data becomes very different from the true/observed distribution? For example, how is the bound in Theorem 3.3 derived? What is the proof idea of Theorem 3.3?

**Limitations:**

- The paper discusses only the potential positive impacts, while no technical limitations are discussed.

**Strengths And Weaknesses:**

### Strengths
- Understanding fair learning from a clustering perspective is novel and interesting.
- The paper is overall well-organized. For example, the diagrams and the presentation of the theoretical results help readers understand the method.
- Theorem 3.3 is interesting and provides a new insight of fairness measure such as DP in view of clustering purity and balance.


### Weaknesses
- Performing GAFC seems similar to existing fair clustering methods [1, 2, 3, 4, 5]. For example, we may sequentially build balls as: (i) first collect samples with similar labels and (ii) then apply an existing fair clustering method to each collection. What would be the main difference between this approach and GAFC? And what would be the main benefits of GAFC compared to this kind of fair clustering-based approaches?
- Only $\Delta DP$ is reported in experiments. Reporting $\Delta LDP$ as well would clarify the practical validity or benefits of the proposed method.
- Although FairGBFC does not perform much worse than existing GB-based methods, I expect that it may require more computation time than other baselines such as f-FERM, Debias, etc. Can the authors also provide a comparison of computation time for these baselines?
- I know additional well-known baselines [6, 7] that can be applied to multi-valued sensitive attributes. Comparing against more baselines would strengthen Table 3.
- The empirical improvement over existing methods seem not very significant, so the practical advantage of the proposed method is somewhat weak.


   [1] https://arxiv.org/abs/1802.05733

   [2] https://proceedings.mlr.press/v97/backurs19a.html

   [3] https://arxiv.org/abs/1906.08207

   [4] https://arxiv.org/abs/2106.07239

   [5] https://openaccess.thecvf.com/content_CVPR_2020/papers/Li_Deep_Fair_Clustering_for_Visual_Learning_CVPR_2020_paper.pdf

   [6] https://proceedings.mlr.press/v80/agarwal18a/agarwal18a.pdf

   [7] https://proceedings.mlr.press/v139/celis21a/celis21a.pdf

---

> ### Author Rebuttal · Authors · 2026-03-31
>
> We thank the reviewer for the constructive feedback. Our point-by-point responses are as follows:
>
> **Q1:** What would be the main difference between existing fair clustering methods and GAFC? And what would be the main benefits of GAFC compared to fair clustering-based approaches?
>
> **Response:** GAFC differs from those methods by performing adaptive joint optimization in GBs rather than applying static constraints. While standard global parity distorts local manifolds, GAFC adaptively calibrates a local fairness penalty ($\lambda_j$) based on each GB's geometry and demographic imbalance. It avoids the feature space distortion of two-stage partitioning by jointly optimizing compactness and fairness. This preserves natural data distributions while ensuring fairness across multiple granularities.
>
> **Q2:** Reporting ∆LDP as well would clarify the practical validity or benefits of the proposed method.
>
> **Response:** For $\Delta\text{LDP}$ (Table below), FairGBFC achieves near-zero scores. Baselines yield N/A on Default as balls contain only a single sensitive group; FairGBFC (0.008) maintains diversity. Thus, Default is omitted due to incomputable baselines.
>
> |Dataset|GBKNN|ACC-GBKNN|GBKNN+|FairGBFC|
> |-|-|-|-|-|
> |Approval|0.033|0.137|0.137|**0.001**|
> |Law|0.193|0.012|0.012|**0.000**|
> |Por|0.259|0.256|0.121|**0.006**|
> |German|0.325|0.371|0.115|**0.003**|
> |Adult|0.062|0.147|0.121|**0.013**|
>
> **Q3:** Can the authors also provide a comparison of computation time for other baselines such as f-FERM, Debias?
>
> **Response:** We have evaluated the average computation times (in seconds) for the baselines, as shown in the table below:
>
> |Dataset|CFA|$f$-FERM|Unbias|GDebias|DP\_PP|
> |-|-|-|-|-|-|
> |Approval|2.303|7.450|3.300|0.314|0.176|
> |Law|47.742|4.532|85.861|5.122|3.036|
> |Por|1.249|4.275|2.112|0.139|0.129|
> |German|2.253|3.769|4.614|0.211|0.208|
> |Adult|31.559|5.081|199.471|8.996|6.118|
> |Default|71.551|10.257|132.199|9.032|14.956|
>
> FairGBFC (Table 1) is the fastest method across all datasets, notably 759x faster on Law. On large-scale Adult/Default, it is 1.2x–1.24x faster than runners-up, proving superior practical scalability.
>
> **Q4:** How does FairGBFC compare against established baselines [6,7] for multi-valued sensitive attributes?
>
> **Response:** As requested, we evaluated Reduction [6] (results in the following table):
>
> |Dataset|Acc|F1|$\Delta\text{DP}$|$\Delta\text{EO}$|
> |-|-|-|-|-|
> |Law (Race)|0.949|0.974|0.010|0.029|
> |Adult (Race)|0.803|0.476|0.128|0.516|
>
> Reduction achieves comparable accuracy, but its $\Delta\text{EO}$ on Adult remains high. FairGBFC (Table 3) outperforms it in both fairness and utility.
>
> **Q5**: The empirical improvement over existing methods seem not very significant, so the practical advantage of the proposed method is somewhat weak.
>
> **Response:** FairGBFC's practical advantages extend beyond marginal gains via three core strengths:
> 1) Improvements: FairGBFC breaks accuracy-fairness trade-offs. On Por, $\Delta\text{DP}$/$\Delta\text{EO}$ drop >60%/80% vs. runner-up. Furthermore, it achieves significant performance in both metrics across all other datasets. On Adult-Race, it hits top accuracy while cutting bias by >30%/50% vs. second-best.
> 2) Efficiency: Faster than all other fairness methods.
> 3) Interpretability: It makes transparent geometric decisions based on quantifiable properties, which is crucial for high-stakes deployment.
>
> **Q6:** However, does this step distort the observed ratio of each group? Can the authors explain behind the gap between perturbing the true ratio of each group and the theoretical results? How can the proposed method still achieve fairness when the distribution of the augmented data becomes very different from the observed distribution? How is the bound in Theorem 3.3 derived? What is the proof idea of Theorem 3.3?
>
> **Response:**
> 1) Distortion: Modifying $P(S=v)$ to $1/N_S$ corrects historical bias rather than distorting it; by preserving original label rates, the augmentation step keeps conditional distributions $P(Y|S)$ and $P(X,Y|S)$ unaltered, maintaining the intrinsic data manifold.
> 2) Theoretical gap: Equalizing global $P(S)$ is mathematically required for our theoretical bound. Imbalance causes GBs to miss groups ($m_j < N_S$), exploding the "missing group penalty" in Theorem 3.3, rendering the bound ineffective. Augmentation bridges this gap by ensuring $m_j\to N_S$.
> 3) Fairness under shift: Augmentation preserves $P(Y|X,S)$ to build fair geometric boundaries without manifold shifts. During inference, true test samples fall into these geometrically fair GBs, robustly inheriting their spatial fairness.
> 4) Theorem 3.3 proof idea: In Appendix E, we first decompose global $\Delta\text{DP}$ into local label allocation differences across GBs. These are bounded by structural imbalance ($\epsilon_j$) and missing group penalty $(N_S-m_j)N_{min,j}$. Summing these yields Theorem 3.3, proving the necessity of augmentation ($m_j \to N_S$) and GAFC ($\epsilon_j\to0$).

---

> > ### Author Rebuttal · Reviewer_7fBk · 2026-04-03
> >
> > Thank you for the rebuttal. While a few parts are still not completely clear to me, for example, about the relation to existing fair clustering and the augmentation argument, I think the rebuttal addressed most of my concerns. Therefore, I will retain the score.

---

> > > ### Author Response · Authors · 2026-04-03
> > >
> > > Thank you for acknowledging that our previous rebuttal addressed most of your concerns. We appreciate your candid feedback that the relationship between GAFC and existing fair clustering methods, and the logic of our data augmentation, remains unclear.
> > >
> > > We apologize for the lack of detail previously. The strict first-round word limit compelled us to dedicate space to new experimental results, compressing these theoretical discussions. To eliminate any ambiguity, we provide a thorough response below clarifying these core aspects, which we hope will fully resolve your remaining reservations.
> > >
> > > **Response to Q1:**
> > > 1) Pipeline Failure: The reviewer's suggested pipeline is intuitive, but our empirical tests show it causes severe "clustering failures." Local collections are naturally skewed. Existing methods enforce rigid global parity constraints. Forcing them into skewed micro-regions makes them aggressively pull distant samples to meet these constraints, destroying the collected label purity and compact geometry, or causing non-convergence. Crucially, this fails even with our data augmentation. Augmentation equalizes global populations, but micro-regions remain naturally heterogeneous during recursive splitting. Rigid global methods still crash locally.
> > > 2) GAFC’s Core: GAFC is a localized splitting algorithm that abandons rigid global quotas. It dynamically calculates an adaptive local penalty ($\lambda_j$) for each individual ball based on its specific internal geometric size and local imbalance degree. This softly guides fair splitting without breaking natural geometry or label purity, ensuring the successful division of highly skewed small balls.
> > > 3) Differences from [1-5]: Unlike [1, 3, 4], which enforce strict micro-assignments that shatter continuous spherical boundaries, GAFC uses smooth joint EM optimization to maintain perfect geometric shapes. Furthermore, unlike [2], which applies a globally fixed penalty ($\lambda$) that over-penalizes small balls and under-penalizes large ones, GAFC solves this by computing a dynamic $\lambda_j$ locally. Finally, unlike [5], which alters data features via neural networks and destroys the physical interpretability of Granular Balls, GAFC achieves fairness strictly through shifting boundaries in the original feature space.
> > >
> > > In summary, GAFC is specifically tailored for our proposed fair Granular Ball generation. Applying existing methods [1-5] locally causes splitting failures during the partitioning process, which prematurely halts the generation and ultimately produces impure granular balls. GAFC’s adaptive $\lambda_j$ uniquely solves this, enabling smooth, fair recursive splitting without geometric distortion or purity loss.
> > >
> > >
> > > **Response to Q6:**
> > > 1) Distortion vs. Correction & Fairness under Shift: Modifying the global marginal $P(S)$ to $1/N_S$ corrects historical sampling bias rather than distorting the truth. Crucially, our augmentation strictly preserves the conditional distributions $P(Y|S)$ and $P(X,Y|S)$. The intrinsic data manifold and feature-label relationships remain strictly unaltered; we merely equalize group densities. Regarding how fairness holds under shift, because $P(X,Y|S)$ is preserved, our GBs act as geometric decision boundaries perfectly aligned with the true manifold. During inference, test samples from the true, skewed distribution fall into these geometrically fair GBs. Since GAFC ensures each GB treats all groups equally locally, this spatial fairness robustly guarantees fairness on the true distribution.
> > > 2) The Theoretical Gap & Necessity of Augmentation: Equalizing global $P(S)$ is mathematically required to constrain the local fairness bound. Consider a highly imbalanced scenario (e.g., Group A:10,000, Group B:10). If the dataset is partitioned into 50 GBs, most balls will inevitably miss Group B entirely ($m_j<N_S$). This inevitable absence causes the "missing group penalty" in Theorem 3.3 to explode, breaking the theoretical guarantee. Augmentation bridges this gap by ensuring sufficient minority samples, making it feasible for GAFC to allocate all groups into every ball ($m_j\to N_S$).
> > > 3) Proof Idea of Theorem 3.3: The proof (detailed in Appendix E) bridges global $\Delta\text{DP}$ with local geometric properties via three logical steps. First, we decompose the global $\Delta$DP by expressing it as the sum of local label allocation differences ($\sum |n_{j,u} - n_{j,v}|$) across all positive GBs. Second, using the triangle inequality, we bound the disparity within any single GB by two geometric terms: the structural imbalance $\epsilon_j$ (excess samples) and the missing group penalty $(N_S-m_j)N_{min,j}$. Finally, summing these local bounds across all GBs yields Theorem 3.3. This derivation proves exactly why our framework requires both Augmentation (to eliminate the missing group penalty by ensuring $m_j \to N_S$) and GAFC (to minimize structural imbalance $\epsilon_j\to 0$).

---

### Official Review · Reviewer_qbvo · 2026-03-15

**Soundness:** 3
**Presentation:** 2
**Significance:** 2
**Originality:** 3
**Overall Recommendation:** 3
**Confidence:** 3

**Summary:**

This paper addresses the prevalent group unfairness problem in data-driven classifiers by proposing a fair classification method based on granular-ball computation (FairGBFC). Existing methods (preprocessing, processing, and postprocessing) often face challenges such as poor interpretability or distortion of the original data distribution.
To address this, the authors propose a two-stage framework: Stage 1 (FairGBG): First, a neighborhood mixup strategy is used for data augmentation to balance the distribution of sensitive groups. Then, a group-aware fair clustering algorithm (GAFC) is proposed, recursively generating "granular-balls" with high purity and balanced internal distribution of sensitive attributes. Stage 2 (FairGBFC): In the classification inference stage, based on the purity and structural balance of the generated granular-balls, an adaptive label assignment mechanism is designed to achieve classification that balances accuracy and fairness.

**Compliance With Llm Reviewing Policy:**

Affirmed.

**Key Questions For Authors:**

The primary problem the authors focus on is addressing group fairness in data-driven classifiers, particularly mitigating the systematic biases these models exhibit toward certain demographic groups in high-risk domains.Specifically, the authors aim to solve the key challenges and limitations that persist in existing fairness-aware classification methods:Limited Interpretability: Many current methods rely on complex model constraints during the training phase (in-processing), which often suffer from poor interpretability.Data Distortion and Utility Loss: Pre-processing and post-processing techniques typically modify feature distributions or prediction outcomes to achieve fairness. This risks distorting the intrinsic data structure and compromising the overall utility and accuracy of the model.Scalability Issues: Existing approaches that rely on global-level interventions struggle to scale efficiently when applied to large datasets.

**Limitations:**

Here are the main limitations of the proposed FairGBFC framework:Heavy Reliance on Synthetic Data Augmentation: The framework depends heavily on the "Neighborhood Mixup" strategy to artificially balance the sensitive groups before generating the granular balls. The ablation studies explicitly show that without this augmentation step, the model's fairness metrics ($\Delta DP$ and $\Delta EO$) degrade significantly. This is a critical limitation because synthesizing artificial data points is often strictly prohibited or considered unreliable in highly regulated, high-stakes domains like healthcare, criminal justice, or credit scoring.Vulnerability to the Curse of Dimensionality: The Granular-Ball Computing (GBC) mechanism relies on geometric distance metrics (specifically Euclidean distance) to partition the data and measure cluster compactness. While this works well for the relatively low-dimensional tabular datasets tested in the paper (e.g., Adult, German, Law), Euclidean distance becomes increasingly meaningless in high-dimensional feature spaces. The framework's scalability and effectiveness on ultra-high-dimensional data, such as raw images or deep text embeddings, are unproven and likely to suffer.Narrow Scope of Fairness Defined: The theoretical guarantees and optimization objectives of the proposed Group Aware Fair Clustering (GAFC) algorithm are strictly confined to Group Fairness (specifically Demographic Parity and Equalized Odds). The framework does not account for Individual Fairness (ensuring that two fundamentally similar individuals receive the same prediction) or Intersectional Fairness (mitigating bias against subgroups defined by multiple overlapping sensitive attributes, such as race and gender simultaneously).

**Strengths And Weaknesses:**

Solid Theoretical Foundation: The paper goes beyond heuristic algorithms, providing rigorous theoretical boundary proofs. Theorem 3.2 demonstrates that increasing the purity of the grains effectively lowers the theoretical upper limit of Local Inequity (LDP). Theorem 3.3 further links global Demographic Parity (DP) with the structural imbalance of local grains, providing theoretical support for the entire framework design. Good Interpretability: Unlike the black-box constraints commonly used in deep neural networks, this method utilizes the inherent geometric topology of GBCs for decision-making. Each decision region (grain) has quantifiable purity and structural balance indices, making the fairness intervention process of the model highly transparent. Detailed Experimental Validation: The authors conducted a detailed evaluation on six widely used fairness benchmark datasets (such as Adult, Law, German, etc.). Experimental results show that FairGBFC maintains competitiveness in classification accuracy and F1 score. Meanwhile, compared to other fairness algorithms based on multilayer perceptrons, this method also performs well in terms of inference efficiency (Time).

Key Weaknesses and Improvement Suggestions: Strong Dependence on Data Augmentation (Mixup): Ablation experiments (Fig. 4 and Fig. 5) show that the fairness metric of the model drops sharply without the Neighborhood Mixup module. Suggestion:This strong dependence on the distribution of preprocessed data may be limiting in high-risk domains where generating false data is strictly prohibited (such as medical or specific judicial scenarios). The authors should further discuss the semantic rationale of Mixup-generated features in real-world contexts, or explore whether GAFC alone can achieve acceptable baseline fairness without altering the global data distribution. Insufficient Discussion on Scalability for High-Dimensional Data:In extremely high-dimensional datasets (such as feature spaces containing image or long text embeddings), Euclidean distance-based granular-sphere partitioning often suffers from the "curse of dimensionality." Suggestion: The benchmark datasets used in this paper are mostly low-dimensional tabular data. The authors are advised to supplement or discuss scalability strategies for this method in higher-dimensional feature spaces. Limitations of Fairness Metrics: This study primarily focuses on group fairness, particularly LDP and DP. Recommendation: In light of recent research trends, individual fairness and intersectional fairness (which considers a combination of multiple sensitive attributes) have also received considerable attention.

---

> ### Author Rebuttal · Authors · 2026-03-31
>
> We sincerely thank the reviewer for appreciating our solid theoretical foundation, good interpretability, and detailed experimental validation. Below, we address your concerns and suggestions in detail:
>
> **Q1: Strong dependence on Data Augmentation (Mixup) in high-risk domains.**
>
> **Response:** We agree that generating 'false data' in high-stakes domains is highly sensitive. We mitigate this through our Mixup's semantic rationale and GAFC's robust standalone performance:
> 1) Semantic Rationale of Mixup: Our strategy avoids hallucinating out-of-distribution records. Interpolation is strictly confined to a seed sample and its $k$-nearest neighbors within the same sensitive group. Semantically, this acts as a continuous, localized oversampling technique—conceptually akin to SMOTE, which is widely accepted and utilized even in medical diagnostics for imbalanced rare diseases. Crucially, we strictly preserve the original positive label rate $P(Y|S)$ for each group (**Sec. 3.1.1** in the manuscript), mathematically guaranteeing semantic integrity without distorting the underlying distributions.
> 2) Baseline Effectiveness of GAFC Alone: While **Fig. 5** in the manuscript shows fairness degrades without augmentation, this is only relative to FairGBFC. Against SOTA baselines (**Tables 2 and 3** in the manuscript), GAFC alone remains competitive. For example, on the Por dataset, GAFC alone yields $\Delta\text{DP} \approx$ 0.026 and $\Delta \text{EO} \approx$ 0.055, vastly outperforming baselines like $f$-FERM ($\Delta\text{DP}$ 0.101, $\Delta \text{EO}$ 0.193). Even on the challenging Adult-Race dataset, GAFC maintains a baseline $\Delta \text{EO} \approx$ 0.31.
>
> As detailed in Appendix **Table 5** in the manuscript, some datasets suffer from extreme demographic imbalance. This severe gap in sensitive group proportions naturally limits the performance of GAFC without augmentation. Crucially, this empirical phenomenon perfectly corroborates Theorems 3.2 and 3.3: without global equalization, granular balls inevitably lack extremely scarce groups, causing the "missing group penalty" to inflate the theoretical upper bound of unfairness. Furthermore, to explicitly verify the indispensability of Granular-Ball (GB) partitioning, we added an experiment applying our augmentation strategy directly to standard classifiers (please refer to our response to Reviewer 4, Q1).
>
> **Q2: Insufficient discussion of scalability and vulnerability to the curse of dimensionality on high-dimensional data.**
>
> **Response:** We agree that Euclidean distance faces the "curse of dimensionality" in high-dimensional spaces (e.g., images and text). However, this work strictly targets tabular data—the prevalent format in high-stakes domains (such as finance and law)—where our framework is highly scalable. GB generation avoids $\mathcal{O}(N^2)$ pairwise calculations, computing distances only to ball centers. For future scaling to ultra-high-dimensional data, two strategies are viable:
> 1) Projecting raw inputs into lower-dimensional latent spaces via deep encoders before ball construction.
> 2) Switching the distance metric from Euclidean to cosine similarity, which is robust in high dimensions.
>
> We thank the reviewer for highlighting this promising trajectory.
>
> **Q3: Limitations of fairness metrics (e.g., lack of individual or intersectional fairness).**
>
> **Response:** We acknowledge our method does not address individual fairness, as this requires fundamentally redesigning the underlying geometric distance and similarity metrics. However, our framework natively supports intersectional fairness. To demonstrate this, we added the COMPAS dataset and evaluated Race and Sex as a joint sensitive attribute on COMPAS and Adult. The results (best results are in bold, second-best are marked with †) are shown below:
>
> | Dataset (Race x Sex) | Method |Acc|F1|$\Delta\text{DP}$| $\Delta \text{EO}$ |
> | :--| :---| :---| :---| :---|:---|
> | **COMPAS** | FairGBFC | **0.686±0.004** | 0.705±0.019† | **0.178±0.073** | **0.202±0.053** |
> | | $f$-FERM | 0.679±0.008† | **0.709±0.008** | 0.342±0.033 | 0.401±0.086 |
> | | Unbias | 0.660±0.010 | 0.694±0.024 | 0.228±0.122† | 0.281±0.123† |
> | | DP_PP | 0.669±0.014 | 0.694±0.020 | 0.365±0.030 | 0.387±0.063 |
> | **Adult** | FairGBFC | 0.812±0.006† | 0.538±0.040† | **0.140±0.028** | **0.236±0.098** |
> | | $f$-FERM | **0.816±0.002** | **0.590±0.008** | 0.292±0.014 | 0.700±0.163 |
> | | Unbias | 0.790±0.010 | 0.403±0.082 | 0.164±0.035† | 0.675±0.161† |
> | | DP_PP | 0.792±0.005 | 0.479±0.035 | 0.210±0.050 | 0.691±0.202 |
>
> FairGBFC achieves a vastly superior accuracy-fairness trade-off under intersectional constraints. On COMPAS, it yields the lowest fairness errors and highest accuracy. On Adult, while baselines suffer from severe intersectional bias, FairGBFC drastically suppresses $\Delta \text{EO}$ to 0.236 and $\Delta\text{DP}$ to 0.140. This proves that our GB mechanism effectively mitigates bias across overlapping minority subgroups.

---

### Official Review · Reviewer_gw6q · 2026-03-18

**Soundness:** 2
**Presentation:** 3
**Significance:** 2
**Originality:** 2
**Overall Recommendation:** 3
**Confidence:** 3

**Summary:**

This paper introduces FairGB, an interpretable fairness-aware classification framework that leverages Granular-Ball Computing (GBC) to embed fairness directly into the data partitioning process. The authors claim that standard bias mitigation techniques often rely on complex constraints or data modifications that can distort data or obscure decision-making, and decomposes the framework in two main stages: Fair Granular-Ball Generation (FairGBG) and Fair Granular-Ball-based Fair Classification (FairGBFC). In the FairGBG stage, a Neighborhood Mixup strategy globally balances sensitive groups, and a novel Group Aware Fair Clustering (GAFC) algorithm recursively partitions the data into granular-balls (GBs) while enforcing strict local demographic equilibrium. During inference, the FairGBFC stage dynamically prioritizes these GBs based on their internal purity and structural balance to assign predicted labels.

**Compliance With Llm Reviewing Policy:**

Affirmed.

**Key Questions For Authors:**

Please refer to the Strengths And Weaknesses section.

**Limitations:**

N/A.

**Strengths And Weaknesses:**

Soundness: overall the paper is theoretically rigor. and the authors provides solid mathematical foundations for the fairness guarantees. The experiments are well-designed, with six well-known fairness benchmarks, and the baselines are well-chosen, encompassing both standard Granular-Ball (GB) methods and state-of-the-art fairness interventions across pre-processing, in-processing, and post-processing paradigms.

Presentation: the overall presentation is good and logically organized.

Significance and originality: the idea of using Granular-Ball generation for bias mitigation is interesting. While fairness has been extensively studied in neural networks and decision trees, introducing group fairness directly into the multi-granularity data partitioning process of GBC is novel, and the proposed method demonstrates a superior trade-off between Accuracy/F1 and DP/EOd compared to baselines. However, I still have a few conerns regarding the overall methodology:

1. What is the motivation of applying Granular-Ball generation for fair classification? Although the authors provide theoretical guarantees for the methodology, it is unclear the advantages of choosing GB over alternative clustering methods.

2. Does the proposed method gernealize to high-dimensional complex data? What is the potential computational cost?

3. Any discussion on the tightness of fairness bounds in Theorem 3.2 and 3.3?

---

> ### Author Rebuttal · Authors · 2026-03-30
>
> We sincerely thank the reviewer for appreciating the theoretical rigor, well-designed experiments, and the novelty of introducing group fairness directly into the Granular-Ball (GB) partitioning process. Below, we address your concerns in detail:
>
> **Q1: What is the motivation of applying Granular-Ball generation for fair classification?**
>
> **Response:** We thank the reviewer for this insightful question. Rather than treating GB generation and clustering as mutually exclusive, we utilize clustering as the core splitting engine within our supervised GB framework. While our GAFC method ensures demographic equilibrium within clusters, purely unsupervised clustering cannot independently guarantee label-dependent fairness. GB generation bridges this gap by recursively splitting data via GAFC until a strict label purity threshold ($P_j \ge \tau$) is satisfied (**Sec. 3.1.3** in the manuscript). Crucially, Theorem 3.2 establishes that enforcing high purity directly suppresses the theoretical upper bound of local unfairness ($\Delta\text{LDP}$), which unsupervised clustering cannot achieve natively. Furthermore, FairGBFC leverages these quantifiable internal characteristics during inference (Eq. 8), prioritizing regions that maximize $P_j \cdot \exp(-\epsilon_j)$ to ensure decisions are dominated by highly reliable and structurally balanced granular regions.
>
> **Q2: Does the proposed method generalize to high-dimensional complex data? What is the potential computational cost?**
>
> **Response:** Our method is highly scalable and generalizes efficiently, supported by both theoretical complexity and empirical results. The computational cost of our framework is highly efficient during both the generation and inference stages:
> 1) Training Complexity (FairGBG): As established in Theorem B.2 (**Appendix B.2** in the manuscript), our foundational algorithm for GB partitioning, GAFC, achieves a linear complexity of $\mathcal{O}(N \cdot K \cdot d)$ per iteration. While our manuscript explicitly detailed the time complexity of the GAFC algorithm, we welcome this opportunity to clarify the time complexity of our complete framework. Since FairGBG recursively applies GAFC with a fixed $K=2$, the parameter $K$ acts as a constant and is consequently omitted from the Big-$\mathcal{O}$ notation. At any tree depth, the total samples across all GBs sum to at most $N$, capping the per-level computational cost at $\mathcal{O}(N \cdot d)$. Assuming relatively balanced splits, the recursive tree depth is $\mathcal{O}(\log N)$, yielding an overall generation complexity of $\mathcal{O}(N d \log N)$. This guarantees near-linear scaling with both sample size ($N$) and feature dimensions ($d$).
> 2) Inference Complexity (FairGBFC): As detailed in **Section 3.2.1**, inference is highly efficient. A test query evaluates distances only to the $N_{GB}$ GB centers rather than the entire training set, strictly bounding the complexity to $\mathcal{O}(N_{GB} \cdot d)$ per sample. Because the data manifold is highly compressed ($N_{GB} \ll N$), FairGBFC achieves massive speedups. Furthermore, unlike traditional fairness methods, we completely bypass complex model training by directly assigning labels based on the intrinsic properties of these compact balls (Eq. 8).
>
> As shown in **Table 1**, FairGBFC requires minimal execution time on large datasets (e.g., Adult, Default). In stark contrast, traditional fairness baselines predominantly train complex models, incurring massive computational overhead. For a detailed comparison of computation times, please refer to our response to Reviewer 3 (Q3).
>
> **Q3: Any discussion on the tightness of fairness bounds in Theorem 3.2 and 3.3?**
>
> **Response:** The fairness bounds in Theorems 3.2 and 3.3 are not loose theoretical artifacts; rather, they are tight under worst-case scenarios and serve as exact, actionable optimization targets for our framework.
> 1)	Tightness of Theorem 3.2: This bound is theoretically tight. In the worst-case scenario—where all minority label samples within a GB are concentrated entirely in the smallest sensitive group—the actual $\Delta\text{LDP} (gb_j)$ exactly equals the upper bound. Our framework actively compresses this bound toward zero by enforcing a strict purity threshold ($P_j\ge \tau$) during recursive splitting and minimizing structural imbalance ($\epsilon_j \to 0$) via GAFC, thereby achieving tightness in practice.
> 2)	Tightness of Theorem 3.3: Tightness heavily depends on the missing-group penalty $(N_S-m_j) N_{min,j}$. By employing Neighborhood Mixup to enforce demographic balance prior to splitting, we prevent missing groups ($m_j \to N_S$). This completely eliminates the penalty, tightly shrinking the global bound to $\frac{N_S}{N} \sum \epsilon_j$, which GAFC subsequently drives to near zero.
>
> Across benchmarks, measured fairness metrics approach zero alongside these theoretical limits, confirming our bounds act as tight, practical optimization guides without loose overestimation.

---

### Decision · Program_Chairs · 2026-04-30

**Decision:**

Accept (regular)

**Comment:**

Two reviewers provide weak reject scores, while one gives weak accept and one gives strong accept. The positive reviewers indicate that most concerns have been addressed in the rebuttal.

The rebuttal provides detailed responses to the concerns raised by the weaker reviews, although no formal acknowledgments were posted. Notably, the weaker reviews also recognize the novelty and potential value of the approach.

Given the overall balance of reviews, I recommend weak accept.